# Current Trends in Cancer Nanotheranostics: Metallic, Polymeric, and Lipid-Based Systems

**DOI:** 10.3390/pharmaceutics11010022

**Published:** 2019-01-08

**Authors:** Catarina Oliveira Silva, Jacinta Oliveira Pinho, Joana Margarida Lopes, António J. Almeida, Maria Manuela Gaspar, Catarina Reis

**Affiliations:** 1iMedUlisboa, Faculty of Pharmacy, Universidade de Lisboa, Av. Prof. Gama Pinto, 1649-003 Lisboa, Portugal; catarina.m.oliveira.silva@gmail.com (C.O.S.); jopinho@ff.ulisboa.pt (J.O.P.); aalmeida@ff.ulisboa.pt (A.J.A.); 2Faculty of Pharmacy, Universidade de Lisboa, Av. Prof. Gama Pinto, 1649-003 Lisboa, Portugal; joanamargaridalopes@campus.ul.pt; 3IBEB, Faculty of Sciences, Universidade de Lisboa, Campo Grande, 1749-016 Lisboa, Portugal

**Keywords:** nanotheranostics, gold nanoparticles, polymeric nanoparticles, lipid-based nanosystems, cancer treatment, cancer imaging

## Abstract

Theranostics has emerged in recent years to provide an efficient and safer alternative in cancer management. This review presents an updated description of nanotheranostic formulations under development for skin cancer (including melanoma), head and neck, thyroid, breast, gynecologic, prostate, and colon cancers, brain-related cancer, and hepatocellular carcinoma. With this focus, we appraised the clinical advantages and drawbacks of metallic, polymeric, and lipid-based nanosystems, such as low invasiveness, low toxicity to the surrounding healthy tissues, high precision, deeper tissue penetration, and dosage adjustment in a real-time setting. Particularly recognizing the increased complexity and multimodality in this area, multifunctional hybrid nanoparticles, comprising different nanomaterials and functionalized with targeting moieties and/or anticancer drugs, present the best characteristics for theranostics. Several examples, focusing on their design, composition, imaging and treatment modalities, and in vitro and in vivo characterization, are detailed herein. Briefly, all studies followed a common trend in the design of these theranostics modalities, such as the use of materials and/or drugs that share both inherent imaging (e.g., contrast agents) and therapeutic properties (e.g., heating or production reactive oxygen species). This rationale allows one to apparently overcome the heterogeneity, complexity, and harsh conditions of tumor microenvironments, leading to the development of successful targeted therapies.

## 1. Introduction 

Cancer, a complex, heterogeneous, and aggressive disease, is globally recognized as overwhelmingly challenging in terms of clinical management. Recent worldwide data from GLOBOCAN estimate 18.1 million new cases and 9.6 million deaths due to cancer in 2018 [1]. Therefore, a precise diagnosis combined with effective treatments are vital for improved clinical outcomes. In this context, the recent advancements in nanotechnology and combined biomaterials have been providing novel nanosystems with increased complexity and multiple properties to respond to the distinct mechanisms involved in cancer development.

Hybrid nanotheranostic systems are multicomponent individual particles, made of organic and inorganic materials, capable of both diagnostic and therapeutic actions, such as for drug/gene delivery, laser-assisted therapy (e.g., photodynamic therapy (PDT) and photothermal therapy (PTT)), and/or imaging modalities (e.g., magnetic resonance imaging (MRI) and positron emission tomography (PET) [2]. The association of these individual agents to a nanocarrier system expanded and enhanced the available nanotheranostics tools, offering the possibility of integrating both tumor accumulation, drug release, and multiple imaging modalities for optimal treatment outcomes [3]. Often, these nanosystems are functionalized with biocompatible polymeric layers and/or targeting moieties, including contrast agents. Additionally, drug release is frequently controlled by an external stimulus, such as ultrasound, light, thermal, or chemical triggers (e.g., pH alterations), allowing for the succeeding interaction between the imaging and the therapeutic part of nanoparticles [3].

In light of the fast developments in the field of nanotheranostics and the need for regular updates on products’ advances, this review focuses on polymeric, metallic, and lipid-based nanosystems for application on solid tumors (Figure 1).

Hence, for the appraisal of current literature, a non-systematic review was conducted between September and October 2018. We have identified both original research and review articles, reporting recent techniques for the development of nanotheranostics, based on polymeric, metallic, and lipid-based nanoparticles for cancers at early or metastatic stage. Only the most recent and representative publications indexed in PubMed/MEDLINE or Scopus, written in English and/or Portuguese were considered. Otherwise, research articles were manually obtained from previous publications.

## 2. Theranostic Nanosystems: From Single to Multimodality

### 2.1. The Evolution of the Design and Composition of Nanotheranostic Systems

In terms of design, it is generally accepted that nanosystems with a size between 5 and 200 nm (and around 100 nm) are adequate for tumor targeting; nevertheless, this assumption is dependent on the type of tumor and its specificities (e.g., blood flow) [4]. For instance, clusters of nanoparticles with a size of ~100 nm, but with an individual size of 5 nm, were shown to be released when reaching the acidic media of the tumor [3]. Moreover, Dreifuss and co-workers have found different responses concerning the applications foreseeable by the size of the gold nanoparticles, ranging between 20 and 120 nm [5]. In their study, the authors verified that a size that allowed the highest tumor uptake (i.e., optimal drug delivery) was not equal to that capable of achieving the highest contrast enhancement (i.e., optimal tumor image) [5]. In the case of liposomes, it is well-known that size influences loading capacity, as well as stability, interaction with biosurface, and biodistribution profile of the encapsulated drugs. On the one hand, smaller sizes (80–200 nm) are associated with increased stability and extravasation to tumor sites. On the other hand, larger liposomes have more loading efficiency, but may become unstable and are also more easily recognized by RES and, consequently, cleared from blood circulation [6,7,8]. Therefore, the size of the nanoparticles should present a balance between an enhanced drug loading (increased size) and favorable pharmacokinetics (smaller size) [3]. A summary of the features of nanotheranostic systems is displayed in Figure 2.

Other characteristics, mostly related to morphology, surface charge and composition have been described to be highly variable and, in fact, dependent on the proprieties attributed to the nanoparticles (e.g., stability, cell penetration, and toxicity) [10]. Dykman et al. (2016) have categorized the production of multifunctional theranostic nanoparticles in three main routes: hybrid nanoparticles (composed of different nanomaterials, such as metals, biopolymers, and lipids), multifunctional nanoparticles (functionalized with targeting moieties and/or drugs), and multifunctional hybrid nanoparticles (incorporating both characteristics) [11]. With a focus on the latter group, which presents the most promising (and challenging) approaches to theranostics, those nanosystems are generally modified to combine more than one composite for a core-shell structure. Moreover, surface functionalization allows for an active targeting to cancer cells and/or combination of light-based modalities (e.g., PDT and PTT) [11]. During the fabrication of these nanoplatforms, the provision of in vivo stability is of foremost importance, as they will be exposed to drastic conditions within the tumor microenvironment. Therefore, an efficient delivery of drugs or diagnostic markers to biological targets in cancer relies on the structure and optimized formulation of those same multifunctional nanoplatforms.

Regarding the biodistribution, both passively and actively targeted nanoformulations have been studied for cancer imaging and treatment for several years [4]. To achieve a non-invasive monitoring of drug release and tumor targeting, MRI contrast agents, such as gadolinium and manganese, are considered more suitable, since the signal enhancement is dependent on interactions with the surrounding media [4]. Thus, the signal is generated upon the drug release from the nanosystem, for instance, triggered by an external stimulus. Certainly, recent discoveries have led to the creation of systems that allow double or triple simultaneous actions. For instance, Kostevsek et al. (2015) developed a one-step formulation of gold–iron oxide nanoparticles functionalized with chitosan, as a biocompatible polymeric shell, which combined both photothermal heating properties and photoacoustic imaging (PAI) with a contrast agent [12]. Another study focused on the development of a dual-imaging theranostic prodrug system, capable of producing a photo-controlled release of cisplatin and performing simultaneously optical and MRI-based guidance [13]. At a first approach, this gadolinium–platinum complex allowed a real-time assessment of the biodistribution and pharmacokinetics of the anticancer treatment, in addition to an increased retention at the tumor site compared to free cisplatin [13]. Willerding et al. (2016) developed long-circulating thermosensitive liposomes (TSLs) co-encapsulating doxorubicin and a gadolinium-based contrast agent. The combination of heat therapy with TSLs led to a significant improvement on drug delivery at tumor sites, and the in vivo MRI monitoring proved to be advantageous for the assessment of local drug release and heating patterns [14].

As interactions between nanomaterials and biomolecules, tissues (healthy and cancerous cells), and biologic organisms are enormous and complex, integrated nanotheranostic systems can include ways to measure those dynamics, for example, by real-time imaging [15]. Hyaluronic acid (HA) is macromolecule and a polysaccharide that has been extensively studied as a tumor target, since it is a natural ligand for the CD44 receptor overexpressed in many cancers [16]. In addition, as a polymer, HA presents several advantages for functionalization of both polymeric and metallic nanoparticles due to the possible linkage by amine groups, and features such as its improved stability, biodegradability, biocompatibility, low toxicity, and low immunogenicity [17,18,19,20]. Thus, HA was studied as a tripodal conjugate (i.e., targeting, imaging, and therapy), with a cytotoxic agent (gemcitabine) and, separately, a probe (THP, which is a gadolimium chelator) for in vivo imaging by single photon emission computed tomography/computed tomography (SPECT/CT) [20]. Several cancer cell lines were assessed, namely human pancreatic carcinoma, murine colon carcinoma, and both murine and human melanoma, but only the murine colon carcinoma (CT26 cell line) showed significant differences between the conjugate and the drug alone in terms of cytotoxicity. Uptake via CD44-receptor was also dose- and time-dependent in colon and pancreatic cells, but other internalization mechanisms appeared to be involved [20]. In vivo results confirmed the uptake of the fluorescent conjugated, while the HA–gemcitabine conjugate allowed a significant delay in tumor growth after 28 days with four injections at approximately 15 mg/kg [20]. One singularity observed in this work was the selection of individual conjugates with a drug and a probe that were already used in clinics but were administered simultaneously. Hence, an approach like this one could favor the regulatory approval of nanotheranostics, rather than the conjugate comprising both features [20], as seen with other cases of nanotheranostic platforms under clinical assessment and that could benefit from a faster approval and further access to patients. Recently, extensive progress has also been made in the use of poly(lactic-co-glycolic acid) (PLGA)-based hybrid multifunctional nanosystems [21]. In one example, both SPIONs and ICG were encapsulated into a PLGA core for NIFR and MR imaging modalities. This last system was then used for siRNA delivery [22]. In another study, a PLGA-dendrimer hybrid nanoplatform was developed for co-delivery of doxorubicin and paclitaxel [23]. Furthermore, a three-layered nanosystem, composed of a PLGA core, a liposome intermediate layer, and a chitosan outer layer, was used for simultaneous delivery of three chemotherapeutic drugs: doxorubicin, paclitaxel, and silybin [24]. For a synergistic effect, a targeted and temperature-responsive PLGA-based nanosystem (PFH/DOX@PLGA/Fe3O4-FA) combined imaging by high-intensity focused ultrasound (HIFU) and therapy by HIFU and doxorubicin [25]. Finally, copper oxide PLGA nanoparticles have been designed for simultaneous MR and ultrasound imaging and PTT [26].

Recently, the diverse applications and features of light-triggered nanotheranostics for cancer treatment have been reviewed [9]. Aside from other inorganic metallic nanoparticles, gold nanoparticles can be prepared with different geometries, such as nanospheres, nanoshells, nanorods, or nanocages. In addition to inorganic photosensitizers (PSs), cancer phototherapy using organic PSs has also been researched, with some already in clinic [9,27]. However, despite their light-controlled efficacy and biocompatibility, the pharmacokinetics is often poor. One approach to solve this limitation could be the use of multifunctional lipid-based nanosystems. As an example, Spring and colleagues (2016) have designed nanoliposomes incorporating a porphyrin-based photoactivable chromophore, and nanoparticle-based encapsulated therapeutic agent, successfully achieving tumor photodynamic damage, as well as light-triggered chemotherapy [28].

In general, these multifunctional hybrid nanotheranostics are expected to offer additional functions related to imaging and therapeutic features, while maintaining the capabilities of conventional nanoparticles, e.g., drug delivery, an enhanced permeation and retention (EPR) effect, and easy surface modification for coating and functionalization [9,29,30].

Among the described light-triggered nanoparticles [9], hybrid multifunctional systems exploring more complex conjugations of inorganic–organic materials can lead to a modified design and properties and, consequently, the imaging or therapeutic outcomes. Nevertheless, barriers caused by the tumor’s specificities and challenges in clinical translation remain and must be accounted for when developing new delivery techniques and clinically relevant tumor models to evaluate the efficacy and safety of systems. 

### 2.2. Exploring Tumor Microenvironment for Improved Nanotheranostics Targeting 

Delivering an effective treatment to the tumor site is a complicated and demanding task, mainly due to the multiple barriers and players involved in its growth and progression. Researchers have dedicated their efforts to understand and reproduce the tumor microenvironment, aiming at creating the most appropriate and realistic scenario for the action of anticancer therapies. One aspect that has been greatly discussed is the accumulation of nanosystems in solid tumors owing to the EPR effect [3,29,30] (Figure 3). Recognized firstly as a major opportunity to enter the tumor and favor the uptake of nanoparticles, depending mainly on their size, shape, and charge, it is now being questioned due to the complexity and variability of the tumor barriers [11,31,32]. Indeed, intratumoral distribution of nanoparticles is highly variable and it is affected by intrinsic factors, such as interstitial fluid pressure (IFP), blood flow, diffusion, and stroma thickness [3,32]. In addition, tumor microenvironment presents different physico-chemical characteristics compared to normal healthy cells, such as acidic pH, hypoxia, active efflux pumps, hyperthermia, altered redox potential, and overexpressed molecular biomarkers (e.g., oncogenic proteins) [2].

Considering all those intervening factors, recent nanotheranostics formulations follow a similar trend in taking the most advantage by integrating stimuli-responsive agents/lipids and anticancer drugs. For example, a light-responsive graphene was combined with an anticancer drug (doxorubicin) and a pH-sensitive disulfide-bond linked hyaluronic acid to form a nanogel [33]. This nanogel presented many features, such as heating (thermal therapy), chemotherapy, real-time non-invasive imaging, and, finally, light-glutathione-responsive controlled drug release. In this case, graphene itself was photoluminescent and could thus work as an imaging contrast agent, as well as a heat source by laser irradiation. Moreover, this nanogel allowed for glutathione-mediated delivery of doxorubicin, whose release mechanism was based on the existence of higher levels of glutathione (GSH), within the cytosol of tumor cells, induced by oxidative stress, unlike non-tumoral healthy cells [33]. Following the same rationale, other studies have developed similar multifunctional nanoplatforms based on the response to GSH [34,35]. Relating to lipid-based nanosystems, the development of pH-sensitive liposomes takes advantage of the polymorphic phase behavior of unsaturated phosphatidylethanolamine (PE), such as DOPE (dioleoyl phosphatidyl ethanolamine), which forms an inverted hexagonal phase (HII) rather than bilayers [36,37]. The stabilization of liposomes into bilayers is accomplished by using an acid lipid, such as oleic acid (OA), linoleic acid (LA), and CHEMS (cholesteryl hemisuccinate). In the case of CHEMS, its protonation after exposure to the tumor acidic milieu leads to bilayer destabilization and the consequent release of the entrapped cargo [36,37].

Indeed, both IFP and acidic pH are conditioning factors for the delivery of nanoparticles into the tumor target [2]. The stroma matrix, mainly composed of glycosaminoglycans (GAGs), polysaccharides, and fibrous proteins, can also hinder therapeutic actions, by increasing metastasis formation and drug resistance [2]. A nanotheranostic system based on defect-rich clay was developed, combining a pH-sensitive MRI diagnostic tool to detect the tumor tissue and both acid-enhanced PTT and chemotherapy, to eliminate cancerous cells [38]. This triple action allowed for the reduction of the dose administered in vivo, guaranteeing complete tumor elimination, after near-infrared (NIR) range laser irradiation (808 nm) and consequent release of 5-fluorouracil (5-FU), the chemotherapeutic drug [38]. In fact, NIR light (range between 700 and 900 nm) has been the focus of many researchers looking for a better solution within the field of imaging materials, theranostics, and light-triggered nanoparticles, mostly due to clinical advantages such as low invasiveness (especially to the surrounding healthy tissues), high precision, deeper tissue penetration, and dosage adjustment in a real-time setting [9,39]. Another example where stimuli-triggered imaging and therapy were achieved is the development of anti-endothelium growth factor receptor (EGFR)-targeted liposomes loading both doxorubicin and siRNA, as well as iron oxide nanoparticles with a mesoporous silica shell (MNP@mSiO_2_). The pH-response was ensured by loading therapeutic agents using an ammonium bicarbonate gradient. Considering an in vitro assay, this specifically targeted nanosystem was selective toward BxPC3 pancreatic cancer cells, promoting a synergistic therapeutic effect [40]. Several strategies for targeting the tumor microenvironment based on nanoplatforms are promising for cancer theranostic applications leading the way into clinics. This approach can improve tumor accumulation of drugs, enhance overall treatment efficiency, and provide flexible and precise external control of the time, area, and dosage of therapy compared to single therapy models.

### 2.3. Nanotheranostics as an Efficient and Safe Alternative in Clinic

The challenge of an effective treatment relies on the existence of high heterogeneity among tumors and patients and within tumor subpopulations. Currently, the “one-size-feats-all” is no longer acceptable as the cornerstone of cancer treatment. Considering that patients need tailored solutions for managing their disease, nanomedicine and personalized therapies are emerging.

In spite of the promising results of nanotechnology during the first generation of the drug delivery systems, most have failed to arise from the bench to clinics. There is a need for clinically relevant nanotheranostics in early stage diagnosis and treatments for patients with cancer. Multifunctional hybrid nanosystems show many advantages over single core-shell particles, such as real-time monitoring of drug release, biodistribution and accumulation at the target site, increased therapeutic efficacy, and prediction of therapeutic response (including to disease progression and treatment outcome in real-time) [3,4].

Furthermore, nanotheranostics might then assist in treatment planning, the anticipation of therapeutic response, and monitoring, aiming at personalized medicine [41]. Another functionality is to use these systems for guidance through imaging, for example, in a preoperative setting or as optical guiding probes during the surgical resection of breast cancer and melanoma [42], and finally, to potentially enhance the accuracy in the stage of these cancers. To the best of our knowledge, no current nanotheranostic formulation has been approved for translation into clinical practice. Nonetheless, if applied to the correct patients, selected by stratification and subpopulation screening, for adequately receiving the most suitable treatment, nanotheranostics will guarantee the success of cancer treatment for the upcoming years. In this context, the main nanotheranostics platforms currently under clinical appraisal are summarized in Table 1.

## 3. Nanotheranostics in Diagnosis and Treatment of Cancer 

### 3.1. Polymeric and Metallic-Based Nanoparticles

#### 3.1.1. Non-Melanoma Skin Cancer and Cutaneous Melanoma

The most common skin cancers are the non-melanoma types, i.e., basal cell carcinoma (BCC), and squamous cell carcinoma (SCC), which are derived from epidermal keratinocytes and are frequently detected in an early resectable stage [43]. In 2018, the incidence of non-melanoma skin cancer rose to 1,042,056 cases, corresponding to 5.8% of all cancers in the world, with a particularly increased incidence in Australia and New Zealand [1]. Primarily, those types of skin cancers are usually removed by surgery or desiccation, if possible, or otherwise with radiation therapy. Additionally, PDT has been successfully applied in superficial skin cancers, using photosensitizers such as aminolevulinic acid (ALA) and methyl aminolevulinate (MAL) [43]. Still, one major problem concerning this approach is that those tumors occur mainly in chronically sun-exposed areas, such as the face, ears, scalp, and hands; thus, both the physical and emotional impact of excision are significant for patients [44]. Furthermore, the application of photosensitizers in these cases is limited by the low penetration into the tumor tissue due to their poor solubility, in addition to the related phototoxicity, which can last up to several weeks after PDT [43,45].

Within the nanotheranostic field, research is focused on rationally combining imaging and therapeutic modalities, which restrict the treatment locally in the tumor area, while improving the penetration of the drugs and/or photosensitizers. Recently, a “bottom-up” method was developed to overcome these limitations of PDT. Briefly, this strategy involved the nano-assembling of the photosensitizer phthalocyanine and the anticancer drug mitoxantrone, associating both optical and chemotherapeutic actions [45]. In addition, the nanosystem was assessed for its capability of converting light into heat; indeed, a mild temperature elevation was found, but should be carefully explored to be further proposed for a PTT effect. As a first attempt, the targeting action of this system was verified in breast cancer MCF7 cells and colon SW620 cancer cells, which suggested that this strategy can be potentially suitable for treatment of different tumors, such as skin cancer. Another promising characteristic of the nanoparticles was the fast and significant accumulation in the tumor tissue. Although the nanoparticles also spread quickly to other organs (e.g., the liver and lungs) during the first hours (1–2 h), those were rapidly eliminated from the body within 24 h. Overall, this supermolecular nanoplatform was able to accumulate in tumors and emit fluorescence imaging, as well as inhibit cancer growth by a synergistic effect with laser irradiation and chemotherapy [45].

A study combined both PTT and imaging actions with a biocompatible 9 nm nanotheranostic platform made of gadolinium and copper chalcogenides, with a strong absorbance in the NIR range [46]. Bovine serum albumin was used as a biocompatible mediator and as an agent to improve photostability. This nanotheranostic allowed a bimodal imaging-guided PTT, responding to 980 nm laser irradiation for 5 min, with temperatures increasing up to 50 °C at the tumor site, and demonstrated inhibitory effects on tumor growth [46]. In terms of its biodistribution, this nanosystem accumulated mainly in organs of the reticuloendothelial system (RES) (e.g., liver and spleen). An ovarian carcinoma cell line (SK-OV-3) was selected to assess the efficacy of this system, rather than the skin cancer model; nevertheless, due to the broad and flexible modalities, we considered that this technology could be also translated into a therapy for skin cancer.

Among the skin cancers, cutaneous melanoma (or melanoma of the skin) is the most common type of melanoma, arising from complex genetic mutations in melanocytes [47]. The tumor microenvironment in melanoma is also very heterogeneous, with complex vascular networks and immunogenicity, as well as induced acquired resistance to treatments by upregulation of multidrug resistance (MDR) mechanisms [48]. Treatment of cutaneous melanoma has improved over the last several decades, but the survival of patients with advanced disease is still poor [48,49]. The number of melanoma cases has increased over the last years, with an estimated incidence of 287,723 cases in 2018, which corresponds to 1.6% of all cancers [1]. In comparison with non-melanoma skin cancer, which is responsible for 65,155 deaths (0.7%) per year, melanoma mortality reaches 0.6% (60,712 deaths) despite being less frequent [1]. Conventional therapies (e.g., dacarbazine) for melanoma present many limitations, such as reduced target specificity, severe adverse effects, and eventually MDR. Recent targeted therapies with kinase inhibitors to suppress the MAPK pathway downstream, specific for BRAF V600 mutation (e.g., vemurafenib and dabrafenib) or MEK inhibitors (e.g., cobimetinib and trametinib) are currently available for those patients with advanced melanoma. However, those novel treatments cause cutaneous toxicities and show a short-term response due to acquired resistance. On the other side, immunotherapy (e.g., ipilimumab and nivolumab) present also a limited response due to acquired resistance and immune-related side effects [50].

In parallel to the development of more potent and specific therapies, an important step for the success of melanoma treatment is its early detection [51]. When there are no metastases, this cancer can be removed by surgery; however, the risks of intervention and of cancer recurrence must be measured in patients with locally advanced tumors (Stages I–III). In those situations, treatment with high doses of interferon-α is administered to improve chances of disease-free and overall survival. As a new line of research, adjuvant treatments are being explored, comprising immune and target therapies in combination with chemotherapy, in addition to neoadjuvant strategies, applied before surgical removal of the tumor [52,53]. In this setting, nanotheranostics might have great involvement, if capable of reducing the size of the tumor and facilitating an effective surgery, or even avoiding it by complete tumor elimination. Moreover, theranostic nanoparticles can be useful for localizing the tumor at an early stage through imaging techniques or for monitoring the treatment delivery directly to the tumor cells [54].

In recent studies, both tumor necrosis factor (TNF) or epidermal growth factor receptor (EGFR) have been selected for targeting treatment approach, as well as the signal transducer and activator of transcription 3 (STAT3), to enhance the apoptosis of melanoma cells [55]. As a first approach, Labala et al. developed a layer-by-layer chitosan-coated gold nanoparticles for local delivery of imatinib to melanoma cells using STAT3 siRNA as the targeting moiety for the inhibition of melanoma cell growth [56]. These nanoparticles with a size around 150 nm were able to reduce tumor growth in a concentration-dependent way, reducing the protein expression by almost 50% in murine melanoma cell models [56]. Thus, those gold nanoparticles provided the ideal optical imaging and tracking functionalities to produce a theranostic platform.

As previously mentioned, gold nanoparticles have been extensively used for both diagnosis and treatment modalities in melanoma cancer, ideally at an early phase and allowing for localized distribution [55]. Gold nanoparticles are very versatile and can be bioconjugated with biomolecules or drugs, which make them suitable for dealing with distinct molecular pathways, such as those involved in heterogeneous cancers, including melanoma [43]. In the field of nanotheranostics, gold nanosystems can be applied for imaging, detecting circulating tumor cells, and, mostly, for a targeted drug delivery, increasing patients’ overall survival [43]. Previously, our research group designed hybrid gold nanoparticles for a photothermal strategy in cutaneous melanoma targeted therapy, which demonstrated potential to destroy melanoma cells at their initial stage by photoactivation and thermoablation, considering a neoadjuvant setting [18]. Nanoparticles were successfully coated with hyaluronic and oleic acids (HAOA) and conjugated with epidermal growth factor (EGF) as a key relevant medical peptide. HAOA-coated gold nanoparticles show a broad absorbance band instead of a narrow absorbance peak (lower activation energy) around 700 nm, allowing an adequate relation between low activation energy and high depth, which enables the laser to irradiate this superficial cancer completely [18]. In vivo assays showed that those spherical gold nanoparticles were actively internalized by tumor cells through EGFR-mediated endocytosis. Furthermore, the nanoparticles associated with NIR-based photothermal therapy (808 nm laser irradiation for 5 min) reduced tumor volume by 80% and caused several coagulative necrotic foci on tumor tissue, but no significant damage of the surrounding tissue or any other side effects [18]. Additionally, the same gold nanoparticles were directly coupled with another cytotoxic abietane (i.e., 6,7-dehydroroyleanone) [57], and their design and composition were adapted for polymeric nanoparticles loaded with a novel anticancer drug (i.e., parvifloron D), demonstrating also promising results in targeting and reducing the growth of melanoma cells [19]. In summary, our studies allowed us to conclude about the flexibility and adaptability of these systems, in addition to applications such as the conjugation of a photosensitizer, evolving to a triple-action nanotheranostic system.

Trying another approach with PTT, Chen et al. (2018) employed an NIR-absorbing polymer such as sulfonated poly(*N*-phenylglycine), and a supercharged green fluorescent protein (ScGFP) to form a hybrid nanoparticle with imaging-guided therapeutic action [58]. These novel theranostic nanoparticles were able to convert NIR light (808 nm laser irradiation) into local ablation of melanoma cells after 5 min, without damage to the surrounding tissues [58].

As a conclusion, and considering the enormous advantages made in this field, especially in light-triggered systems for PTT, it is possible that soon enough skin cancers, including melanoma of the skin, can be treated with a safe and efficient image-guided and laser-based treatment, focusing on the tumor area and, eventually, eliminating the tumor without the need of surgery.

#### 3.1.2. Head and Neck Cancers

Head and neck cancers represent approximately 4% of all diagnosed cancers [1,59]. Around 90% of these tumors are squamous cell carcinomas, which include tumors of the oral cavity, larynx, pharynx, and glands [1,60]. The incidence of these tumors is approximately 550,000 cases per year worldwide [60,61]. Moreover, according to recent estimates, the burden of head and neck squamous cell carcinoma (HNSCC) is increasing in low income countries, such as those in the Asia-Pacific region and Africa [59,60]. The most relevant risk factors are tobacco smoking, alcohol consumption, and human papillomavirus (HPV) infections [61]. Multidisciplinary standard treatment for these locally advanced tumors includes surgery, postoperative radiotherapy, and/or chemoradiotherapy [62]. Tumor location defines the treatment selection, with tumors in the oral cavity usually being resected by surgery, and oropharynx, nasopharynx, hypopharynx, and laryngeal carcinomas treated with radiation and chemotherapy to preserve the organ structure. Nevertheless, morbidity associated with invasive procedures, incomplete resection, and drug resistance reveals an unmet need for less invasive and safer therapeutics.

Regarding molecular pathways, HNSCC shows an 80–90% overexpression of EGFR, considered as an implicated key target for chemotherapies [63,64]. Associated with EGFR is the signal transducer and activator of transcription 3 (STAT3), constitutively activated in several cancers including HNSCC, and this contributes to uncontrolled tumor growth (via anti-apoptotic mechanism) [65]. As most patients do not respond or develop resistance to current treatments [61], nano-delivery systems can allow higher drug bioavailability, local targeting action, and reduced toxicity to healthy tissues [66,67]. Hybrid nanoparticles made of polymeric, metallic, and other bioactive compounds and functionalized with targeting moieties are being studied to mitigate these limitations.

First described in 2011, AGuIX^®^ products are ultrasmall (size < 5 nm) paramagnectic gadolinium-based nanoparticles with a polysiloxane matrix, providing both contrast agent properties and radiosensitizing efficacy. These nanosystems have been developed to improve MRI diagnosis of solid tumors while providing treatment with radiotherapy and are currently under translation to the clinics [68]. The main advantages of these nanostructures are the high tolerance (few side effects) and renal elimination, as well as the possibility of using different administration routes (intravenous or intratumoral injection). Furthermore, this nanosystem allows for the combination of photosensitizers (e.g., tetraphenylporphyrin derivative) for another therapeutic action throughout PDT. Thus, this nanotheranostic offers a dual optimized therapeutic effect by radiation therapy and light-based photosensitizing modality, which can benefit from MRI properties in the detection of the tumor location and guide the treatment within its borders [68].

In this context, it seems that a new generation of nanotheranostic particles is under study, such as those comprising new metallic materials (with different properties) and surface functionalization with multiple targeting moieties. Among all available nanostructures for light-based applications, gold nanoparticles have been widely studied due to their high absorption coefficient, potential versatility, and functionalization [69]. Recently, a dual theranostic system made of gold nanoparticles was explored to target the activation of STAT3 in head and neck cancer, using a combination of cell-surface nucleolin and radiosensitizing approaches [70]. Nanoparticles were functionalized with the targeting oligonucleotide and were radiolabeled to enhance cancer cell uptake. Internalization of the oligonucleotide gold nanoparticles induced apoptosis via enhanced DNA double-strand break formation. Eventually, these nanosystems can be applied as a primary option as an adjuvant radiation therapy for post-surgery or non-resectable head and neck cancer [70].

Moreover, 20 nm spherical gold nanoparticles coated with glucose and cisplatin were produced, in combination with radiation treatment, to ensure an imaging guided treatment for head and neck cancer [71,72]. d-glucose was previously described as a natural contrast agent for CT scan, MRI, PET, and SPECT [73]. In the work conducted by Davidi and Dreifuss, glucose worked as a radiosensitizer and as a carrier to deliver cisplatin by means of glucose transporter-1 receptor internalization, which was overexpressed in head and neck cancer cell lines (e.g., A431) [71,72]. To guarantee the activity of glucose as a targeting ligand and improve the receptor mediated endocytosis, conjugation with the gold nanoparticles was conducted with glucose’s second carbon atom. Here, a synergistic result was observed in terms of tumor reduction when using the conjugated gold nanoparticles and cisplatin, since this metal can also act as a radiosensitizer. Nevertheless, it was found that, after injection in tumor-bearing mice, the conjugated nanoparticles not only were present in the tumor site and kidneys but also managed to cross the blood–brain barrier (BBB) and reach the brain tissue (about 5.8%) [71,72]. Identical to other approaches, these gold nanoparticles were coated with a natural ligand that fulfills a specific interest in anticancer treatment and potentially improves tumor reduction, compared to the conventional available treatment. However, the effects of the undesired and unspecific accumulation of the therapeutic agents and nanocarriers should be carefully measured fostered by a safer medical treatment.

In addition to polymeric-based hybrid nanoparticles, multifunctional systems can also comprise lipid molecules associated with any inorganic material. As an example, porphyrin (inorganic PTT agent) has been self-assembled with lipid film, made of pyropheophorbide-lipids, cholesterol, and polyetheneglycol (PEG2000-DSPE), forming a nanotheranostic agent called porphysome [74]. This system combines a tracking modality based on PAI and fluorescence imaging allied with PTT treatment (Figure 4). Each imaging technique has a different aim, since the photoacoustic signal can detect the intact nanoparticles present at the tumor site, and the fluorescence imaging can track the delivery of the porphysome. Both features allowed a real-time guidance and dose adjustment of the treatment [74]. In this study, the porphysome showed a tumor-specific accumulation (highest peak after 24 h post-injection) and a general biodistribution to the spleen, liver, and small intestine of a rabbit model with buccal carcinoma. Nonetheless, these nanosystems were able to clearly delineate the tumor margins (a five-fold increased signal) and promote tumor necrosis after irradiation for 100 s (around 1.6 min) at 671 nm. No cellular damage nor any other side effects were observed in other organs (including the heart, liver, spleen, lung, kidney, and salivary gland) [74].

In summary, photoacoustic and other imaging-guided modalities are under assessment to deliver a more precise, efficient, and safe treatment for head and neck cancers. Indeed, PAI have been recognized as an early procedure for starting radiation therapy in patients with head and neck cancer [75] and, thus, providing a real-time imaging of dynamic changes in tumor oxygenation, which influences the necessary dose and the overall efficacy of the radiation treatment. Additionally, the combination of PTT and PAI might promote a synergistic therapeutic effect since the imaging modality also allows optical excitation in the NIR range, promoting a local heating of the tumor tissue [75]. Although clinical use of NIR fluorescence-guided surgery is limited due to a few approved contrast agents, further development of specific fluorescent probes, such as anti-EGFR antibodies, can increase the use of these techniques for diagnosis, treatment, and the follow-up of head and neck cancers [39]. Nanotheranostics can present specific characteristics that confer the specificity needed to treat a subpopulation of tumors; they otherwise might be suitable for many different tumors such as those of breast, prostate, or thyroid cancer. NBTXR3^®^ is an example of a promising nanotheranostic made of hafnium oxide nanoparticles combined with radiation therapy that was first assessed for head and neck cancer and is currently being assessed in clinical trials for other solid cancers, such as rectal cancer, prostate cancer, and breast cancer (Table 1).

#### 3.1.3. Thyroid Cancer 

Thyroid cancer accounts for 1% of all cancers and is the most common endocrine tumor, with 64,300 cases and 1980 deaths in the USA in 2016 [76,77]. The incidence has increased over the past few years, reaching 567,000 cases worldwide in 2018 [1]. Furthermore, this cancer is responsible for 5.1% of the total estimated female cancer burden, with a global incidence three times higher in woman than in men [1]. The survival rate is still quite high, leading to 0.4% of deaths [1,77,78]. This cancer also presents a complex etiology, which is still not well understood, and is associated with multiple risk factors, among which is childhood neck radiation, Hashimoto’s thyroiditis, family history of thyroid adenoma or cancer, familial adenomatous polyposis, obesity, smoking, and hormonal exposures [1,76]. Overall, thyroid cancer arises from follicular cells in the thyroid and is the most common endocrine malignancy [79]. Advances in nuclear technologies and new diagnostic techniques allowed one to improve the success of differentiated diagnose of thyroid cancer, which also led to the increased incidence of this cancer in many countries [1]. Along with these technological contributions, theranostic radioiodine has also emerged for a personalized management based on molecular imaging (used since 1950) and highly effective treatment [79].

In this context, nanotheranostics are gaining space in recent years for applications in optimization of a tailored thyroid cancer medicine. Up until now, several studies reported the use of nanoparticles in the diagnosis and/or treatment of thyroid cancer. Carbon nanoparticles, for example, are widely investigated in this type of tumor [80,81,82,83]. Wang and collaborators evaluated the efficacy of carbon nanoparticles of about 150 nm to be accurately identified in patients undergoing central lymph node dissection surgery [82]. The study aimed at reducing an erroneous excision, as well as evaluating the value of this nanosystem in obtaining a faster recovery of the parathyroid gland function. In fact, the authors concluded that it was possible to delimit the lymph nodes with high precision, therefore allowing the surgery to be as accurate as possible, so that other glands, such as the parathyroid glands, were not affected [82].

Another study evaluated the efficacy of perfluorocarbon nanoparticles as a diagnostic method for thyroid cancer, by ultrasound molecular imaging, and for SHP2-targeted cancer treatment [84]. The nanoparticles were produced by double emulsion, in which trichloromethane was first dissolved in poly(lactic-co-glycolic acid) (PLGA), followed by the addition of perfluorocarbon. The obtained polymeric precipitate was further activated by the addition of carbodiimide crosslinker using EDC/NHS coupling, and the SHP2 antibody solution and polyethylenimine were then added. Finally, DOTA-NHS was conjugated to the nanosystem for biomedical imaging purposes. Results showed that the formulation had a high affinity for thyroid cancer cells due to the overexpression of SHP2 in these tissues compared to healthy thyroid tissue, and could be activated by low-intensity focused ultrasound (LIFU)-triggered radiation to enhance ultrasound molecular imaging, allowing for the identification of the tumor [84].

One study developed silicon dioxide nanoparticles conjugated with a specific ligand for thyroid cancer, the thyroid-stimulating hormone receptor (TSHr), and loaded with a well-known anticancer drug (e.g., doxorubicin) [85]. As previously described, the tumor environment, in which the bond between doxorubicin and cis-aconitic anhydride is broken and the cytotoxic compound can be rapidly released, shows an acidic pH. Therefore, for the production of these nanosystems, doxorubicin was first conjugated with cis-aconitic anhydride, followed by conjugation with PEGylated silicon dioxide nanoparticles [85]. Briefly, the nanoparticles were obtained by SiO2 junction with NH_2_ groups exposed with succinimidyl carboxyl methyl ester (mPEG-NHS) and orthopyridyl disulfide PEG succinimidyl ester (PDP-PEG-NHS). To ensure the specificity of the system, TSH was bound to these disulfide bond pathways. In vitro studies in thyroid cancer cells (FTC-133 cell line) demonstrated a significant increase in toxicity, approximately 7.3 times stronger with this nanoformulation compared to the free drug. Indeed, the IC_50_ value of free doxorubicin was 2.32 μM, whereas that of this formulation was only 0.32 μM. Another strong factor highlighted by the authors was the decrease in cardiotoxicity effects by the nanosystem compared to the free doxorubicin [85].

As an alternative for polymeric nanoparticles, Sun et al. explored the advantages of polymeric photothermal agents instead of carbon- and metal-based compounds for formulation of nanotheranostics, and these advantages include improved optical absorption, photostability, and biocompatibility [86]. In this study, nanoparticles made of narrow band gap D−A conjugated polymer (TBDOPV−DT) were produced for PAI and photothermal therapy, triggered by an NIR laser radiation at 1064 nm (increased tissue penetration capacity). In vitro testing demonstrated that the nanoparticles were efficiently taken up by HepG2 and HeLa cell models, while reducing their viability if treated with combined laser irradiation and TBDOPV−DT nanoparticles. In terms of biodistribution, the nanoparticles showed an early accumulation in both liver and lungs. As expected, the nanosystem showed strong photoacoustic signals, which was also beneficial for photothermal performance (photothermal conversion efficiency of 50%) and thus could completely inhibit the growth of tumors and avoid recurrence within 20 days [86]. Overall, nanotheranostics may unravel new approaches for diagnostic and treatment of different clinical and histological features in thyroid cancer, helping to clarify underlying pathogenesis that are still unclear in some of these tumors.

#### 3.1.4. Breast Cancer

Breast cancer is the second most common cancer in the world and the most frequent in women [1,87]. In the USA, for instance, among females, the mortality rate of breast cancer is only surpassed by the lung cancer mortality rate, it is estimated that one in eight women will develop invasive breast cancer [1,87]. In 2018, more than 330,000 new cases of invasive and non-invasive breast cancer are expected in women [88]. Despite the scientific development that allowed advances in the treatment and detection of the disease in earlier states, more than 40,000 women due to breast cancer are expected to die in 2018 in the USA [87,88]. In men, the number is much lower; however, about 2500 new cases of invasive breast cancer are expected [88].

Breast cancer is a complex and heterogeneous type of cancer, characterized by the occurrence of multiple molecular alterations, which can be used as diagnostic and prognostic markers of the disease [87,88]. The most common cause of hereditary breast cancer is an inherited mutation in the BRCA1 or BRCA2 gene, which are tumor suppressor genes involved in such essential functions as DNA repair [87,88]. There are various biological processes and genetic mutations that lead to the appearance of breast cancer and sensitivity to various drugs, such as hormone receptor, human epidermal growth factor receptor 2 (HER2), EGF, vascular endothelial growth factor (VEGF), mechanistic target of rapamycin (mTOR), and cyclin-dependent kinase 4/6 (CDK4/6) [89]. Women, compared to men, present a 100-fold increased risk of having breast cancer; furthermore, aging and heredity also contribute as predisposing factors to the disease [88]. Many non-genetic risk factors of breast cancer are involved, such as race and ethnicity, benign breast conditions, proliferative breast lesions, lobular carcinoma in situ or lobular neoplasia, chest radiation therapy, exposure to diethylstilbestrol, lifestyle and personal behavior-related risk factors of breast cancer, birth control and contraceptives, hormone replacement therapy after menopause, excessive alcohol consumption, significant overweight or obese, not having children or not breastfeeding, and a lack of physical activity [88].

With the improvement in science and medicine, targeted therapies that increase the efficacy and precision of the treatment and decrease the toxicity are now an ambitious solution under development [89]. Several studies have reported the use of nanoparticles as a strategy to combat breast cancer, such as a new VEGF-targeted nanocarrier, composed of magnetic nanoparticles coated with albumin that carry doxorubicin and in which it conjugates monoclonal antibodies to VEGF, which increases their specificity for the tumor [90]. Additionally, due to its magnetic core, this system can still be used for diagnosis, and its detection is possible by MRI after i.v. injection. The nanoparticles were synthesized by thermal decomposition of iron (III) acetylacetonate in benzyl alcohol, coated with BSA (bovine serum albumin) and PEG and conjugated with anti-VEGF monoclonal antibodies [90]. In this study, researchers optimized the nanoparticle size to less than 50 nm, as well as the compounds present in the coating, since these are two important factors for the nanoparticles to remain longer in the bloodstream. Regarding the coating, nanoparticles were pegylated to guarantee an efficient loading process with doxorubicin, to prevent interaction with the plasma proteins and to avoid their absorption by RES, and managed to obtain superior results compared to clinically approved formulations [90].

In another study, β-lactoglobulin nanoparticles were conjugated with folic acid to allow receptor-mediated endocytosis present in cells, and loaded with doxorubicin, which has been shown to be effective against MCF-7 and MDA-MB-231 [91]. First, a β-lactoglobulin solution was incubated with the previously prepared solution doxorubicin hydrochloride for 30 min, to which acetone was added at a constant rate until it became milky. A glutaraldehyde aqueous solution was then added, and the solution was allowed to stir for a few hours. Conjugation with folic acid was done by adding the nanoparticle suspension to a pre-prepared solution of folic acid followed by 1 h of stirring and purification by centrifugation [91]. This formulation was able to retain doxorubicin allowing non-release of the cytotoxic compound into healthy cells as well as the bloodstream, the release being made at the tumor site in response to the more acidic pH characteristic of the tumor environment. In vitro testing demonstrated that the nanosystem presented a greater toxicity profile compared to free doxorubicin after 72 h of incubation, against breast cancer cell lines [91].

Another recent study was based on the grounds that miRNA-21 can be used as a diagnostic and therapeutic biomarker for breast cancer, so gold nanoparticles functionalized with a chemically modified miRNA-21 were developed [92]. This method aimed at suppressing the function of miRNA-21 present in tumor tissues for the inhibition of cell growth and the death of apoptotic cells, while simultaneously using fluorophore-labeled DNA molecules hybridized with antimiRNA-21 for diagnostic purposes. Briefly, the nanoparticles were produced using the sodium citrate reduction method, followed by the addition of antimRNA-21 with labeled DNA molecules [92]. Regarding metal-based systems for nanotheranostic approaches, spherical silver nanoparticles with fluroglucinol were produced according to a simple, green method [93]. In short, silver nanoparticles were prepared by dissolving fluroglucinol in water followed by 20 min of stirring after which silver nitrate was added to the previous solution. Floroglucinol, a polyphenolic compound present in plants and marine algae, shows multiple therapeutic properties, e.g., antioxidant, anti-inflammatory, antimicrobial, anti-diabetic, anti-allergic, and antiretroviral. These nanosystems, which showed a small size (10–50 nm), were tested against MCF-7 breast cancer cell lines, demonstrating remarkable cytotoxic properties [93].

Similar to gold and silver nanoparticles, Fe_3_O_4_ magnetic nanoparticles present broad optical absorption in the NIR range and increased possibilities for biopolymeric coatings and functionalization. In addition, magnetic nanoparticles have the advantage of potentiating supermagnetic proprieties that work as an MRI contrast agent [94]. Hence, Fe_3_O_4_ magnetic nanoparticles conjugated with methotrexate were developed and further encapsulated in hepatitis B virus core (HBc) protein as a shield and stabilizer for the nanocarrier (Figure 5) [94].

This innovative system was tested in vitro for photothermal stability and irradiation-induced apoptosis, as well as in murine breast cancer 4T1 in BALB/c mice exposed to an 808 nm NIR laser for 5 min. Tumor size reduced in 10 days, comparable to treatment with nanoparticles alone or no treatment at all. MRI capability was also assessed both in vitro and in vivo, showing that this system could guide in the tumor position with great precision [94].

Finally, a carboxyl-modified PEGylated poly skeleton, to which the herceptin antibody was modified to present a HER-2 specific surface, was added to superparamagnetic iron oxide nanoparticles (SPIONs), with perfluorohexane and paclitaxel [95]. These SPIONs were activated by NIR laser and allowed the transformation of laser energy into thermal energy, causing the perfluorohexane to be vaporized and consequently release paclitaxel. Therefore, this nanotheranostic comprises another combination of photothermal therapy and chemotherapy for the treatment of breast cancer [95].

In conclusion, multifunctional nanotheranostics including imaging and light-based therapies have been developed to improve the local action and reduction of the impact of chemotherapy in healthy breast tissue. This technique allows for the reduction of the size of the tumor, alike to melanoma and other operable solid cancers, working as a neoadjuvant therapy.

#### 3.1.5. Prostate Cancer

Prostate cancer is one of the leading tumors in terms of incidence (7.1% of all cancers worldwide) and mortality (3.8% of deaths worldwide), surpassed only by lung and breast cancers [1]. In recent years, both screening and diagnostic techniques for prostate cancer, as well as the clinical applicability of key molecular targets and receptors, such as the prostate-specific membrane antigen (PSMA) and gastrin-releasing peptide receptors (GRPR), have considerably improved [96,97,98]. PSMA is a type II transmembrane protein involved in many cellular functions, such as enzymatic functions, cell migration, survival, and proliferation [97]. PSMA is overexpressed in almost all prostate cancers (90–95%), but is also present in healthy cells from the small intestine, proximal renal tubules, and lacrimal and salivary glands, which limits the dose used in radiation therapy [96,97]. Targeted therapies for PSMA have been studied in prostate cancer, mainly through antibodies and antibodies-drug conjugates [96]. GRP receptors are overexpressed in prostate cancers and are found in the gastrointestinal tract [98]. Interestingly, high upregulation of GRPR is specific to prostate carcinoma and occurs mostly during the early stages of this cancer [99]. With regard to the relevant targeting molecules, the GRPR antagonists show increased advantages over the agonists, such as a favorable pharmacokinetics (longer receptor retention followed by rapid clearance), a higher affinity for those receptors and higher in vivo stability (e.g., 177Lu-NeoBOMB1, 68Ga-NeoBOMB1) [98,99]. In addition, to the targeting moieties, anticancer treatments such as taxanes (e.g., docetaxel and paclitaxel) have been explored in association with several imaging and treatment modalities (e.g., radiation therapy and ultrasound imaging) [100,101,102]

Within this scenario, multimodal nanotheranostics could potentially improve sensitivity and specificity in the diagnosis of prostate cancer, for instance, by combining multiple targeting molecules for an effective treatment only in cancerous cells. Previously, the pre-clinical and clinical use of nanoparticles as potential theranostics, including metallic, polymeric, and lipid-based systems, have been reviewed elsewhere [103]. In fact, research in this field has focused greatly on prostate cancer, aiming at an early and simultaneously diagnosis and treatment.

In a first approach to develop a combined theranostic nanocarrier, nanoemulsions and poly(d,l-lactide-co-glycolide) (PLGA) nanocapsules were produced, entrapping a photosensitizer molecule for PDT (chloro aluminum phtalocyanine) [104]. This molecule was used as a therapeutic agent and for tumor localization by confocal laser scanning microscopy after fluorescence promoted by NIR laser irradiation at 670 nm [104]. Both nanosystems presented a negative surface charge (approximately −40 mV) and a mean size around 200 nm, but nanocapsules were able to internalize the prostate cancer cells and promote phototoxicity more efficiently than nanoemulsions [104]. Other research groups proposed a polymeric nanogel (<200 nm), showing intrinsic photoluminescence and encapsulating a naturally fluorescent anticancer drug (doxorubicin) [105]; on the other hand, multi-layered PLGA nanoparticles (~200 nm) were produced for co-encapsulation of theranostic agents (e.g., curcumin, doxorubicin, hydrochloride, and indocyanine green), with chemotoxicity or phototoxicity profiles and intrinsic fluorescence, for a controlled pH-responsive release [106]. Thus, in these two different approaches, it was demonstrated that multiple agents can be encapsulated into the same nanocarrier, or even the nanocarrier itself can be modified, to achieve both imaging and therapeutic actions.

In line with insights from PSMA-target therapies, researchers have also turned their attention to design PSMA-target theranostic nanocarriers for prostate cancer [107,108]. Folic acid is a natural PSMA-targeting ligand [97] and has therefore been explored for functionalization of drug delivery systems. Flores et al. reported the use of folate-conjugated polymeric nanoparticles to deliver a cytotoxic peptide, CT20p, which reduces cancer cell migration and adhesion, consequently leading to the selective elimination of tumors, expressing chaperonin containing TCP-1 (CCT) [107]. Results demonstrated that cell internalization could be mediated by overexpressed PSMA, due to the absence of folate receptors (PSMA (+) pre-treated PC3 and LNCaP cell lines) and that CT20p allowed for a selective elimination of prostate cancer cells, compared to doxorubicin alone [107]. Another approach using also folate-functionalized nanosystems explored the use of MRI based on SPIONs, to increase the sensitivity and specificity of the diagnostic technique and reduce the toxicity associated with contrast agents [109]. Hence, a norbornene-based magnetic copolymer was synthesized, comprising doxorubicin and a high spin Fe^3+^-terpyridine (Fe-Tpy) complex as a T1 contrast agent. The anticancer drug demonstrated an accelerated release (~80% in 24 h) when exposed to an acidic medium, mimicking the tumor microenvironment but not under physiological conditions (pH 7.4). Additionally, these nanosystems increased drug retention > 65% in prostate cancer DU 145 cells after 10 h. Finally, a low IC_50_ was determined to be 1 μg/mL, based on cytotoxicity assays conducted with the same cell lines, demonstrating the anticancer efficacy of this nanotheranostic system [109]. Despite promising results, further studies should focus on the biodistribution of this novel nanoaggregate, which were not described in the study.

As an alternative, Mangadlao and co-workers developed gold nanoparticles, functionalized with PSMA-1 and loaded with a photosensitizer, Pc4, for fluorescent PDT [108]. Gold nanoparticles presented a size of approximately 20 nm, a spherical design, and an absorbance peak at 680 nm, demonstrating that they can be used to selectively target a PSMA-expressing PC3pip cell model, in both in vitro and in vivo assays, with remission of the tumor 14 days after PDT [108]. Overall, this strategy allows for an image-guided surgery of resectable tumors, as well as the elimination of nonresectable tumor ablation by PDT, without substantial side effects on the surrounding healthy tissue.

Considering particularly the unresectable tumors, a palladium radioisotope (103-Pd), used in clinical practice, was conjugated onto hollow gold nanoparticles (~150 nm) and assessed in vivo as brachytherapy seeds for a 5-week localized radiotherapy of prostate cancer [110]. In this case, the nanoparticles were able to retain the tumor tissue for the entire treatment period, while no side effects nor accumulation were observed for preferential organs, such as liver and spleen [110]. Since no functionalization was applied, this high retention rate was explained mainly by the large size of the particles and the intrinsic EPR effect of gold nanoparticles, which preferentially accumulate at tumor sites via passive targeting. Furthermore, and as seen in other works already described herein, the authors propose taking advantage of the gold nanocarriers as radiosensitizers to enhance the DNA damage by radiation [110].

In another study, small 5 nm gold nanoparticles were conjugated with dithiolated diethylenetriamine pentaacetic acid (DTDTPA), to increase their stability and allow for a combination of CT imaging and radiotherapy [111]. Briefly, these nanoparticles achieved moderate uptake by prostate cells, corresponding to passive accumulation in the tumor tissue, dependent on DTDTPA concentration, which protected the nanoparticles from opsonization and clearance by RES. In terms of therapeutic value, these nanosystems improved survival by 31%, compared to the group of mice receiving only radiation therapy [111].

Despite the multiple advantages of those metallic nanosystems, numerous studies have also explored functionalization for the localized treatment of prostate cancer, in order to improve the bioavailability and target specificity of the nanoparticles. Recently, 40 nm gold nanocages were functionalized with neuropeptide Y (NPY), a ligand involved in the regulation of prostate cancer cell growth, using thiolated PEG [112]. The cubic-shaped gold nanocarriers were assessed as intrinsic probes for imaging and tumor ablation via NIR light-based PTT, showing an absorbance between 802 and 806 nm [112]. Additionally, the nanocages were able to stabilize the helical secondary structure and preserve the binding motif of the NPY, so that the receptor recognition ability was not compromised; in addition, the nanosystems were able to stimulate ERK activation, which indicates an interaction with the receptors from prostate cancer cells [112]. Mainly, the authors have proposed that internalization occurred via two mechanisms: micropinocytosis (for aggregated particles above 300 nm) and clathrin-mediated endocytosis (for individualized nanoparticles). Lastly, in vitro testing showed that combining both the NPY-functionalized gold nanocages and NIR laser irradiation at 800 nm for 20 min decreased tumor cell viability, showing extensive necrosis [112].

Another study focused on the design of a triple multimodal nanotheranostic, taking advantage of the intrinsic fluorescence of porphyrin-based nanosystems and combining both PTT and PDT modalities. This nanoporphyrin-based drug delivery system, which is also an inorganic PTT/PDT agent, could promote thermal ablation by NIR light-triggered activation and the formation of reactive oxygen species (ROS) (Lin2018). In addition, these nanoparticles were loaded with two Hsp90 inhibitors, 17-allylamino-17-demethoxygeldanamycin (17AAG) and 17-(dimethylaminoethylamino)-17-demethoxygeldanamycin (17DMAG), to decrease the levels of pro-survival and angiogenic signaling molecules induced by PTT, so that cancer cells would become more sensitive to this therapy [113]. Similarly, the non-functionalized plain nanoparticles were able to accumulate in the tumor tissue while showing a residual presence in other organs, such as the intestine, liver, spleen, and lungs. In terms of in vivo studies, the 17AAG-loaded nanoporphyrin systems associated with light irradiation at low dose (0.5–1.25 W cm^−2^ for 3 min) were more efficient compared to each individual treatment modality and did not cause any significant side effects [113].

Considering the myriad of systems and strategies assessed for an early detection and treatment of prostate cancer, as well as for other tumors, there is a promising possibility of achieving synergistic effects from combining more than one therapeutic modality or imaging tools without potentiating toxic interactions.

### 3.2. Lipid-Based Nanosystems

Among the diversity of lipid-based nanosystems available today, liposomes are definitely the most well-known and versatile ones due to their unique properties. Liposomes are unilamellar lipid bilayers with an aqueous core and present numerous advantages, namely biocompatibility, biodegradability in terms of their main constituents, low toxicity, and the ability to incorporate both hydrophilic and hydrophobic compounds. Being the most extensively studied and successful lipid-based nanosystem, a variety of liposomal formulations are currently already in biomedical use or under clinical trials [6,114,115,116,117]. Besides their ability to carry a diversity of chemotherapeutic compounds, they have also been explored as delivery systems of a great variety of diagnostic agents, including ^64^Cu [118] and ^14^C isotopes [119], quantum dots (QDs) [120], gadolinium (Gd)-based contrast agents [117], SPIONs [121], and fluorescent probes [117,122,123,124]. Taking all these factors into consideration, liposomes emerge as a highly promising theranostic tool, with a broad spectrum of clinical applications in cancer management.

Several successful proof-of-concept applications of lipid-based nanosystems have been described in literature. In the following sections, we will provide several examples of liposomes as theranostic nanosystems for different malignancies. In a separate section, preclinical examples with other lipid-based systems, such as SLNs, LNCs, micelles, and lipid-nanocapsules, will also be addressed. The most relevant data are summarized in Table 2.

### 3.3. Liposomes

#### 3.3.1. Melanoma and Non-Melanoma Skin Cancer

QDs are nanoscale semiconductor crystals that possess adjustable optical properties for biomedical imaging [158,159], presenting size-dependent light emission with appropriate chemical and photostability [159]. In the field of nanotheranostics, QDs are emerging as tools with high potential for such applications, demonstrated by several reports in the domains of drug delivery and biomedical imaging [159]. To maintain their biological activity and minimize possible toxic effects, QDs may be encapsulated in nanoparticles, such as liposomes, for optimal results. Using a xenograft murine melanoma model, Al-Jamal and colleagues [119] performed biodistribution studies including tumor accumulation of liposomes sterically stabilized with DSPE-PEG and encapsulating PEG-lipid coated quantum dots (*f*-QDs) as imaging agent. As expected, the inclusion of DSPE-PEG at the liposome surface dramatically increased blood circulation times, compared with cationic hybrid vesicles. The authors used two main phospholipids for liposome preparation, dioleoyl phosphatidyl choline (DOPC) and distearoyl phosphatidyl choline (DSPC), a fluid and a rigid phospholipid, respectively. Although no differences in terms of blood circulation times were observed, the more rigid liposomal formulation was able to accumulate longer times and to a greater extent at tumor sites. Overall, these liposomal platforms showed excellent potential for future applications in cancer theranostics by co-encapsulating a cytotoxic agent together with QDs.

Glucocorticoids (GCs) exert anti-inflammatory and immunomodulatory effects, and are currently being used in clinics for cancer treatment together with chemotherapeutics [160] to improve the antitumor activity of drugs and to reduce or prevent unwanted side effects [161]. The glucocorticoid prednisolone phosphate (PLP) was co-loaded with a gadolinium-based contrast agent [Gd-DOTAMA(C_18_)_2_] in PEG long-circulating liposomes. This nanosystem allowed for a successful and non-invasive visualization of biodistribution and evaluation of tumor progression in a xenograft melanoma murine model. Importantly, the presence of the contrast agent in liposomal bilayer did not influence drug efficacy, rendering this nanosystem a promising theranostic tool [125].

Using a gadolinium-based MRI agent, the clinically approved gadoteridol, Rizzitelli and collaborators [126] designed long-circulating liposomes and promoted the local release of gadoteridol at tumor sites through the application of an ultrasound-based technique. In further studies, preliminary in vitro results revealed that doxorubicin and gadoteridol presented the same liposomal release profile when co-loaded in the same liposome. As such, the combination of both compounds in one nanosystem might prove to be an advantageous approach for monitoring the effectiveness of in vivo drug release at target sites.

Recently, the potential of long-circulating liposomes functionalized with riboflavin (RF) and loaded with indocyanine green (ICG) for PAI was explored [127]. PAI is an appealing method that offers high resolution within suitable imaging depths, translating light-induced ultrasonographic (US) images to tissue optical absorption [162]. In PAI, the contrast can be obtained from endogenous (hemoglobin, melanin, DNA, and RNA, among others) or exogenous (gold nanoparticles or organic NIR dyes) absorbers. Among the NIR ones, ICG is a clinically approved dye and one of the most vastly researched [163]. RF modulates cell growth and development, mostly required by metabolic active cells, such as tumor cells. Therefore, RF targeting represents an emerging strategy for tumor contrast enhancement and cell-specific drug delivery [127,164,165]. The developed lipid-based nanosystem [127] showed high tumor accumulation, in accordance with previously reported data on ICG-loaded liposomes [128,129,166]. Additionally, it led to increased blood circulation time, improved stability, and enhanced in vivo optical properties, demonstrating the potential for theranostic applications in cancer management.

#### 3.3.2. Breast Cancer

There are evidences indicating that the increased activity and abnormal localization of several cysteine cathepsins within the tumor microenvironment plays a key role in tumor progression [167]. Therefore, the modulation of cysteine cathepsins emerges as a potential antitumor strategy. Mikhaylov et al. [130] prepared long-circulating liposomes that load ferrimagnetic iron oxide (FMIO) nanoparticles to target the tumor and the surrounding microenvironment. Furthermore, a cathepsin protease inhibitor was combined with the liposomes, resulting in a dramatic reduction of tumor growth. This theranostic nanoplatform provided an enhanced MRI contrast and effectively accumulated at tumor sites and the adjacent stroma. The authors believe that this nanosystem could function as a non-invasive and real-time imaging tool with enhanced sensitivity for the simultaneous detection and therapy of breast cancer.

Taking advantage of the solid tumor microenvironment particularities, namely hypoxia, long-circulating liposomes co-encapsulating a hypoxia-activated prodrug, AQ4N and the photosensitizer hexadecylamine conjugated chlorin e6 (*h*Ce6), have been developed [131]. After chelating with ^64^Cu isotope, liposomes displayed properties suitable for positron emission tomography (PET). For the in vivo proof-of-concept, a triple imaging modality combining PET, fluorescence and PAI was assessed in combination with PDT. The results demonstrated the ability of the nanosystem to accumulate at tumor sites, remarkably enhancing the inhibition of tumor growth via a synergistic effect attained by sequential PDT and hypoxia-activated chemotherapy. Overall, this lipid-based theranostic tool represents a promising candidate for future applications [131].

The research group of Rizzitelli [132,133] developed long-circulating liposomes loading both gadoteridol and the chemotherapeutic drug doxorubicin. After systemic injection, they successfully achieved a local tumor drug release from liposomes by ultrasound. This strategy significantly improved the therapeutic efficacy in a breast tumor model, demonstrated by almost complete tumor regression. Despite the promising results, the researchers highlight the need for an evaluation of the protocol in non-superficial tumor models. Nevertheless, they also emphasize two important aspects: (1) the liposomal formulation is quite similar to the clinically approved Doxil^®^; (2) Gadoteridol is also approved for clinical use. In this context, there is a high potential for progression into clinical trials.

Folate receptor (FR) is known to be overexpressed in the cell surface of malignant cells [168,169], allowing a tumor-specific targeting and subsequently a localized imaging and delivery of therapeutic agents [169]. The work conducted by Ma et al. [134] aimed at developing doxorubicin loaded into FR-targeted long-circulating liposomes embedding conjugated polymer dots. These are a class of fluorescent macromolecules that present distinct photochemical and electroluminescence features, including light harvesting properties, high brightness, and good biocompatibility. The modality of fluorescence imaging has been researched for its safety, low costs, and high temporal resolution [170]. The developed liposomal system proved to be a safe and effective platform for the simultaneous therapy and imaging of breast cancer, improving the drug release profile, cytotoxicity, cellular uptake, as well as tumor accumulation [134].

In another study, a different targeting moiety was used: gonadorelin, a peptide analogue of luteinizing hormone-releasing hormone (LHRH), with a high affinity toward the LHRH receptors, which are abnormally expressed in a variety of tumors [135]. Taking advantage of this fact, He et al. prepared gonadorelin-functionalized liposomes carrying both magnetic iron oxide nanoparticles (MIONs) and the conventional chemotherapeutic mitoxantrone. In the proof-of-concept, the use of this nanosystem, integrating both therapeutic and imaging modalities, effectively inhibited tumor progression, while providing a real-time and non-invasive visualization of the therapeutic protocol [135].

Dai et al. [136] further explored another target, integrin α3. This receptor is highly expressed on the surface of the triple negative breast cancer (TNBC) cell line MDA-MB-231, and extracellularly accessible. Therefore, the authors selected it to evaluate the theranostic potential of long-circulating liposomes loading both doxorubicin and an NIR probe (DiD) and functionalized with a cyclic octapeptide for targeting integrin α3. In a TNBC in vivo model, the authors successfully accomplished tumor site accumulation of liposomes and, subsequently, significant tumor growth inhibition and non-invasive real-time monitoring [136]. Moreover, Lozano and coworkers [128] also employed specific targeting for their theranostics strategy. They constructed long-circulating liposomes functionalized with a monoclonal antibody (hCTM01). Both ICG and doxorubicin were loaded onto liposomes. In breast cancer and colon adenocarcinoma models, the authors demonstrated the multifunctionality of the nanosystem, with simultaneous non-invasive and high-resolution tumor imaging by multispectral optoacoustic tomography (MSOT), and chemotherapeutic activity [128].

In the current and future era of personalized medicine, nanotheranostics might be an invaluable tool to select the more adequate treatment regimens, to anticipate therapeutic responses and monitor the patients’ clinical evolution [41]. An example of research focusing on this area was a study conducted in a triple negative breast cancer (TNBC) xenograft model. Among breast cancer, TNBC is one of the most aggressive and therapy-resistant forms of breast cancer [171]. In a recent work, the approach was to personalize the patient’s treatment by firstly probing the tumor sensitivity toward chemotherapeutic agents [137]. The authors constructed long-circulating liposomes loading doxorubicin, gemcitabine, or cisplatin, as well as the corresponding DNA barcode as a screening probe. After systemic administration in a triple negative breast cancer (TNBC) xenograft model, the nanosystem was able to target tumor cells, confirmed by the fluorescent monitoring of the diagnostic imaging agent ICG. Furthermore, the DNA barcode analysis established a correlation between the therapeutic efficiency of each drug, where gemcitabine was the most effective one. Finally, to verify the accuracy of this diagnosis strategy, the free drugs were administered intravenously in the xenograft model where, as predicted, gemcitabine exerted a superior therapeutic effect, compared to the other tested drugs. However, this strategy requires a biopsy, an invasive technique. Nevertheless, the researchers suggest the potential application of this nanoplatform for personalized and improved treatment protocols [137].

PDT is an alternative treatment modality that is attracting interest due to its selectivity, residual systemic toxicity, and non-invasive nature [2]. In an in vivo proof-of-concept of TNBC, long-circulating thermosensitive liposomes were evaluated as a theranostic tool [129]. PDT was the therapy modality by incorporating ICG in liposomes and NIR as detection method. The real-time monitoring showed liposomal tumor accumulation and excellent tolerability, with a nearly complete tumor eradication. The authors emphasized the therapeutic advantages of a liposomal-based nanocarrier against TNBC, and combining it with a non-invasive detection method [129]. In another research work, long-circulating thermosensitive liposomes co-encapsulating doxorubicin and ICG, using NIR as detection method, were designed [138]. In vivo, this remarkable theranostic nanosystem allowed a thermo-controlled release of doxorubicin and a real-time monitoring of its distribution. A complete eradication of tumors with no associated toxic side effects was observed.

Metal-based compounds have been researched for many years as anticancer agents, preventing tumor cell proliferation and inducing DNA damage. The most well-known and the first clinically approved metal-based chemotherapeutic drug was cisplatin [172]. However, several adverse events are associated with this drug, so alternatives are continuously being investigated. Compared to cisplatin, ruthenium (Ru)-based compounds display limited toxic effects [124], and, interestingly, some Ru-derived complexes are more effective against metastases compared to primary tumors [173]. Moreover, Ru-based agents can also function as imaging agents [124]. Long-circulating liposomes incorporating a Ru polypyridine complex were assembled by Shen et al. [124] for the simultaneous fluorescence monitoring and therapy by tumor cell apoptosis. In a mouse model of TNBC, the treatment with this nanosystem dramatically reduced tumor growth in the absence of toxic side effects. According to the authors, this nanoplatform constitutes a biocompatible theranostic with high potential.

As already stated, a tumor microenvironment is complex, presenting several particularities. One of these particular features is the increased H_2_O_2_ production rate and impaired redox balance [174]. Taking advantage of this fact, Chen et al. [139] developed long-circulating H_2_O_2_-responsive liposomes for PAI and PTT. In breast, glioma, and lymph node metastasis tumor models, the researchers verified that this lipid-based nanosystem was sensitive to the H_2_O_2_ present in the tumor microenvironment, enabling an accurate tumor visualization as well as metastasis identification. Moreover, the nanosystem provided the opportunity for PTT by NIR absorbance, constituting a potential theranostic tool [139].

Another distinct feature of solid tumor microenvironment is, as already mentioned, the acidic pH [2,175]. Recently, Zheng et al. [140] explored the use of a tumor-specific and pH-responsive peptide in long-circulating liposomes loaded with both SPIONs and paclitaxel. In a xenograft breast cancer model, the authors were able to successfully monitor by MRI the in vivo tumor progression and confirm the significant tumor growth impairment, evidencing the potential of the lipid-based nanosystem for theranostic applications [140].

#### 3.3.3. Gynecologic Cancer

Among all available therapeutic options for cancer, small interfering RNAs (siRNA) are recognized as highly versatile and potent gene silencers. Despite this, some difficulties associated with siRNA therapy include poor cell uptake, low stability under physiological conditions, off-target effects, and possible immunogenic reactions [176]. A viable strategy to overcome these problems is to associate the siRNAs to cationic liposome systems. Following this line of thought, Kenny and collaborators [141] prepared long-circulating liposomes combining the delivery of therapeutic siRNA with in vivo monitoring of tumor growth evolution, using a gadolinium-based MRI agent. They showed the accumulation and co-localization of liposomes and siRNA at ovary tumor sites. In addition, the intravenous administration of these liposomes led to a significant reduction in tumor growth progression when compared with controls, rendering this lipid-based nanosystem a potentially valuable theranostic tool for cancer applications [141].

Multimodal theranostic nanosystems may include, in the same platform, different therapeutic and imaging properties. For example, Liu and coworkers [142] developed methotrexate-coated long-circulating liposomes loading iron oxide nanoparticles (for MRI and magnet targeting), doxorubicin and the photosensitizer zinc phthalocyanine (ZnPc; for NIR imaging and photodynamic therapy). Methotrexate is a folic acid (FA) analogue, acting simultaneously as a chemotherapeutic drug and as a tumor targeting ligand. The researchers successfully achieved a versatile liposomal platform that, in addition to providing multimodal imaging by fluorescence and MRI, combines chemo and phototherapy in the same nanosystem to ensure synergistic and improved anticancer activity [142].

Adopting a similar strategy, Guo et al. [143] constructed thermosensitive long-circulating liposomes loading oleic acid-modified magnetic nanoparticles and doxorubicin. Liposomal surface was functionalized with methotrexate for a specific targeting to FR overexpressed in cervical tumor cells. The active targeting, combining light and magnetic properties, allowed a local drug release, promoting a synergistic effect in terms of enhanced cytotoxic activity with reduced side effects, compared to free doxorubicin. Importantly, the biodistribution was monitored in real-time by both fluorescence and MRI imaging modalities [143].

Overall, these multifunctional liposomes show great potential for cancer theranostics.

#### 3.3.4. Colon Cancer

Intratumoral (IT) injections of chemotherapies are not commonly applied due to the well-known specific features of solid tumors, namely high IFP and increased density of extracellular matrix (ECM) that impair drug release and its homogeneous distribution [3,32]. To overcome these hurdles, Miranda and collaborators [144] developed long-circulating liposomes, including in the lipid composition porphyrin-phospholipid (PoP), an NIR-responsive lipid for phototherapy. Liposomes entrapped three different hydrophilic cargos: sulforhodamine B (SRB,) a fluorescent dye, gadolinium-gadopentetic (Gd-DTPA), an MRI agent, and oxaliplatin, a chemotherapeutic agent. According to the researchers, this lipid-based nanosystem proved to be advantageous, improving imaging and therapy, following IT injection, when compared to current modalities in clinical use.

#### 3.3.5. Hepatocellular Carcinoma 

Globally, hepatocellular carcinoma (HCC) has been reported as the second leading cause of cancer-related deaths. The currently available therapeutic strategies frequently lead to disappointing survival rates, mainly due to a late diagnosis, disease recurrence, and metastasis [177]. Therefore, there is an urgent need for the research and development of novel and effective options.

In this sense, Shao and colleagues [145] focused on the design of FR-targeted long-circulating liposomes containing a suicide gene system and NIR QDs. After the systemic administration of this nanosystem, researchers successfully achieved an enhanced gene delivery to the target site and were able to monitor, in real-time, its accumulation at tumor sites. Moreover, tumor growth was strongly inhibited, with minimal side effects. These promising results led to the development of a lipid-based theranostic for application in liver cancer [145]. Furthermore, in a xenograft model of HCC, long-circulating liposomes incorporating SPIONs, the stimuli-responsive anethole dithiolethione (ADT), and the pro-drug hydrogen sulfide (H_2_S) were evaluated [146]. An external magnetic field was applied to ensure preferential targeting to tumor sites, allowing the conversion of nanosized liposomes to microsized H_2_S bubbles. Subsequently, they were exposed to high acoustic intensity for tumor tissue ablation, resulting in significant tumor growth inhibition. Importantly, this elaborated and dynamic process was monitored in real-time by both MRI and ultrasound, holding promise as a multimodal imaging nanosystem for cancer therapy [146]. Lee and collaborators [147] obtained long-circulating FR-targeted liposomes loading a gadolinium (Gd^3+^)-texaphyrin core, conjugated to a doxorubicin prodrug. This complex was designed to undergo cleavage in the presence of glutathione (GSH), which is usually highly expressed in cancer cells. In vivo, this dual-mode imaging (MRI and fluorescence) and therapeutic nanosystem provided an accurate and sensitive monitoring of metastasis progression and reduced tumor burden.

#### 3.3.6. Brain-Related Cancer

Glioma tumors derive from glia or glial precursor cells and are recognized as the most prevalent brain tumor. Grade IV glioblastoma multiforme (GBM) is the predominant and most lethal form of glioma, associated with extremely poor prognosis [178]. In 2005, it was reported that the combination of radiotherapy and temozolomide-based chemotherapy resulted only in a 2.5 month increase in the median survival of GBM patients [179] and has been implemented as the standard of care [180].

For targeting glioma tumors, Wu and team [148] prepared liposomes integrating NIR-triggered trimodal imaging modalities (CT, MRI, and fluorescence) and photothermal therapy. The researchers demonstrated that, in vivo, this lipid-based platform led to a complete inhibition of glioma tumor growth, indicating the high potential of its use as a successful theranostic tool [148]. In a recent research work, Xu and coworkers [149] developed long-circulating liposomes loaded with SPIONs, an integrin antagonist, cilengitide, and QDs for simultaneous MRI/NIR imaging and treatment of glioblastoma. In vivo, the nanoplatform successfully crossed the BBB and, upon exposure to an exogenous magnetic field, was able to preferentially accumulate at tumor sites, significantly inhibiting tumor growth. Therefore, this promising liposomal-based theranostic strategy could be applied to ensure an accurate surgery resection of glioblastoma [149].

The study of de Smet et al. [150] was based on the construction of long-circulating temperature-sensitive liposomes (TSLs) incorporating doxorubicin and a gadolinium-based MRI agent [Gd(HPDO_3_A)(H_2_O)] for monitoring of ultrasound-mediated drug delivery. In vivo, the local drug release at gliosarcoma sites was clearly enhanced by ultrasound-induced hyperthermia, and the authors highlight the importance of the imaging feature of these TSLs for an accurate visualization and control of the treatment protocol [150].

Specifically intending to treat central nervous system (CNS) lymphoma, liposomes containing SPIONs and functionalized with rituximab, a monoclonal antibody against CD20, were developed [151]. Additionally, this system was coated with the surfactant Tween 80. In an in vitro model of the BBB, this lipid platform was able to cross it and exert an antitumor effect. In addition, preliminary in vivo studies have established that, after systemic administration, this lipid-based nanoplatform can accumulate at brain lymphoma sites, holding great promise for future applications in patients burdened with this disease [151].

### 3.4. Liposomes in Clinical Studies

In a clinical setting, it is crucial to monitor in real-time and understand how the patient’s condition is evolving and how the treatment protocol can be adapted for optimal outcomes. As each patient and cancer itself present an intrinsic heterogeneity, the use of nanotheranostic tools can help one achieve the full potential of therapies [181]. In this subsection, we have summarized the most recent studies on cancer patients using liposomes for nanotheranostic applications, depicted on Table 3.

### 3.5. Others

Aside from the well-known liposomes, other lipid-based nanosystems have also been explored as nanotheranostics. In the following sections, we will provide some examples of research work conducted on these diverse tools with potential theranostic applications in cancer. Herein, alternative lipid-based systems are described.

#### 3.5.1. Lipid Nanoparticles 

Apolipoprotein E3 (apoE3) nanoparticles have been shown to cross the BBB via transcytosis [185]. This active transport into the brain is thought to occur, in part, by apoE3 binding to the low-density lipoprotein receptor (LDLR) expressed in brain endothelium [186]. Therefore, opportunities for developing glioblastoma-targeted drug delivery nanocarriers arise. Exploring apoE3 as a targeting moiety, porphyrin-lipid nanoparticles were constructed [152]. This nanosystem combined the active targeting toward glioblastoma cells and the therapy and imaging through porphyrins. These molecules display both fluorescent and photodynamic properties, conferring a theranostic modality to the lipid nanoparticles. In vivo, a selective uptake of the targeted nanoparticles by the malignant cells was observed. Overall, apoE3-targeted porphyrin-lipid nanoparticles represent a promising therapeutic and contrasting platform for glioblastoma [152].

Lin and his research team [187] created lipid nanoparticles functionalized with high density lipoprotein (HDL), mimicking the scavenger receptor class B type I (SR-BI), which is up-regulated in several tumors. Overall, the specifically targeted nanoparticle consisted in an NIR fluorescent core enveloped by a phospholipid monolayer, which, in turn, was intercalated with cholesterol-linked siRNA as therapeutic agent. In an orthotopic model of prostate cancer, the researchers concluded that the NIR imaging modality was highly sensitive, allowing a non-invasive and real-time monitoring of drug delivery, as well as treatment response. This versatile and biocompatible platform could potentially be adapted for the delivery of siRNA, chemotherapeutics, or even the combination of both, rendering it a promising theranostic tool with practical applications in cancer management [187].

#### 3.5.2. Solid Lipid Nanoparticles 

Solid lipid nanoparticles (SLNs) are colloidal nanocarriers that have a size range between 50 and 100 nm, generally spherical in shape, being their structure composed of a solid lipid core (at both room and body temperatures), stabilized by an interfacial surfactant layer [188,189]. Several lipids can be used for the preparation of SLNs, including fatty acids, highly purified triglycerides, complex glyceride mixtures, or waxes. As these are physiological lipids, SLNs are biocompatible and present a low toxic potential. In addition, SLNs can enhance drug bioavailability, promote specific targeting, and ensure the stability of the entrapped compounds [190,191,192]. Kuang and coworkers [153] developed targeted SLNs to tumor angiogenic vessels, encapsulating the IR-780 iodide dye to monitor PTT by NIR imaging. The nanosystem was biocompatible, stable in physiological conditions and specifically accumulated at glioblastoma tumor tissue. This versatile platform could be used for targeted and NIR-guided PTT cancer treatment [153]. In addition, the ability of pulmonary-delivered SLN as an imaging tool for gamma-scintigraphy biodistribution studies has also been emphasized, with obvious diagnostic applications. Pre-clinical studies confirmed their potential as a powerful drug carrier for paclitaxel in the treatment of lung metastases [193,194,195].

#### 3.5.3. Nanostructured Lipid Carriers 

Nanostructured lipid carriers (NLCs) are composed of a binary mixture of solid and liquid lipids, forming distinct nanostructures that, in the same way as SLNs, are solid at both room and body temperatures. The lipid matrix of NLCs has a particular nature, which can vary from an imperfect crystallization to an amorphous structure. This feature grants NLCs with higher drug incorporation capacity and release properties, when compared with SLNs [192,196].

Li and his team [154] aimed at producing CXCR4-targeted NLCs loaded with the NIR dye IR780, combining PTT and imaging modalities. The authors successfully achieved a simple, stable and multifunctional nanosystem able to hamper tumor progression and prevent metastasis development. Taking these results into consideration, scientists believe that this lipid-based nanosystem has the potential to be used in a clinical setting [154]. In another study, NLCs co-loading QDs and paclitaxel were developed [155]. In a hepatocellular carcinoma mice model, this nanoformulation improved the antitumor effect of paclitaxel and, at the same time, allowed tumor detection and imaging, devoid of toxicity. The researchers concluded that this approach may set the basis for a successful theranostic nanoplatform [155].

#### 3.5.4. Lipid Nanocapsules

Balzeau and collaborators [156] have designed functionalized LNCs carrying paclitaxel and the far-red fluorochrome DiD for glioblastoma theranostics. These specifically targeted nanocapsules were able to promote their uptake by tumor cells and, in vivo, preferentially accumulated at tumor sites and successfully impaired its progression. The results show that this interesting nanoplatform, combining the modalities of therapy and in vivo monitoring, might represent a step forward in the development of innovative and powerful theranostic tools for cancer treatment [156].

#### 3.5.5. Lipid-Based Micelles

Ma and coworkers [157] created a lipid-based nanomicelle system, exhibiting both therapeutic and imaging modalities. In a xenograft breast cancer model, docetaxel was used as a therapeutic drug, and non-invasive optical monitoring was achieved by bioluminescence and fluorescence imaging. This nanosystem proved to be stable in physiological conditions and to effectively and significantly inhibit tumor growth, with low systemic toxicity. Globally, these promising results might establish the grounds for new advances in the cancer theranostics field [157].

## 4. Meeting the Criteria for a Successful Translation of Nanotheranostics into Clinic Settings

The nanotheranostic field has been progressing over the last several years, but many difficulties still limit its clinical translation. Successful application depends on choosing the most suitable imaging and contrast modalities for the right clinical condition. Thus, an interdisciplinary approach should be strictly followed.

First, regulatory agencies should update the guidelines for these nanotheranostic platforms in order to facilitate the assessment of efficacy and safety and, ultimately, their introduction into clinical use.

Next, for those who research and develop these nanotools, it might be challenging to simultaneously optimize dose levels and administration frequencies using a single delivery platform. In addition, when designing a nanotheranostic formulation, caution needs to be taken so that neither of the components becomes prematurely released from the delivery system. Drug encapsulation optimization, ligand conjugation efficiency, and high reproducibility with low cost biomaterials are essential issues for clinical application of nanotheranostics.

For most nanotheranostics developed thus far, safety in humans has not yet been completely studied. Nanoparticles should be structurally well-defined, reproducibly synthesizable, and derived primarily from biodegradable and/or biocompatible materials. A deeper understanding of the mechanisms by which nanotheranostic are cleared from the body and interact with the immune system is crucial.

Moreover, substantial in vivo assessment is of significance and highly required. Finally, further development is needed to permit high-resolution imaging and to mitigate the background of common techniques such as fluorescence as well as some limitations of photobleaching.

In Table 4, different nanotheranostics reviewed here are compared in terms of key criteria to assess their potential translation into clinical settings.

## 5. Conclusions

In this review, we describe the most representative developments within the field of nanotheranostics that address a precocious diagnostic and treatment of superficial and solid tumors, at an early stage. For those types of tumors explored, there is a tremendous focus on designing multifunctional theranostic nanoplatforms that comprise targeting molecules, photosensitizers, and anticancer drugs under clinical use, in order to improve the time until regulatory approval and clinical translation of these nanotechnologies. However, when we compared it to products that have already been in clinical practice for many years, such as Doxil™/Caelyx™ or Abraxane™, nanotheranostics still show a limited presence in clinical trials. Most difficulties are related to the complexity and synergistic effects resulting from the combined therapeutic and imaging modalities and materials such as those present in the multifunctional hybrid nanosystems.

Thus, great efforts have been conducted by the scientific research community to translate these new nanotheranostics into clinical trials. Overall, for all types of cancers appraised in this review, major concerns are now focused on the optimization of tumor accumulation/retention and biodistribution of both nanoparticles and delivered drugs by imaging techniques, and on learning how these nanomaterials interact with biological systems. We noticed that they are differences regarding the active and passive transportation between them, which could be related to the material used for fabrication of the nanoplatform and the intrinsic interaction with the tumor microenvironment. For this, non-invasive imaging techniques are strongly needed to assess specificity receptor binding and internalization mechanisms (e.g., endocytosis) of the nanosystems into the tumor cells, but a bright future for polymeric, metallic, and lipid-based nanosystems combining diagnostic and therapeutic functions is expected. To achieve this goal, a deep understanding of their interaction, including a safety assessment, must be accomplished considering a biological perspective.

This investigation may open to the development of successful theranostic nanoplatforms in the near future. The interest of pharmaceutical companies to conduct clinical trials and further introduce novel nanotheranostics into the market will be crucial for its success.

As we discussed herein, the role of nanotheranostics can be appreciated at multiple levels, especially with respect to cancer therapy. Formulations of polymeric and metallic nanoparticles and liposomes play a very important role in improving the quality of clinical care and treatments, including predictive, preventive, and personalized medicine. Nanotheranostics may offer the right drug with the right dose to the right patient at the right time. In a very near future, this research should certainly attract an increasing amount of interest from pharmaceutical companies for the development of successful theranostic nanoplatforms with the subsequent introduction of those novel nanotheranostics into the market.

## Figures and Tables

**Figure 1 pharmaceutics-11-00022-f001:**
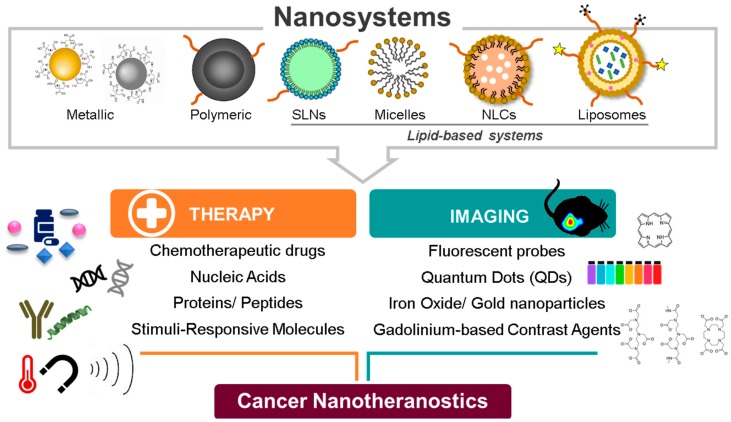
Nanotheranostics: polymeric, metallic, and lipid-based nanosystems for cancer management.

**Figure 2 pharmaceutics-11-00022-f002:**
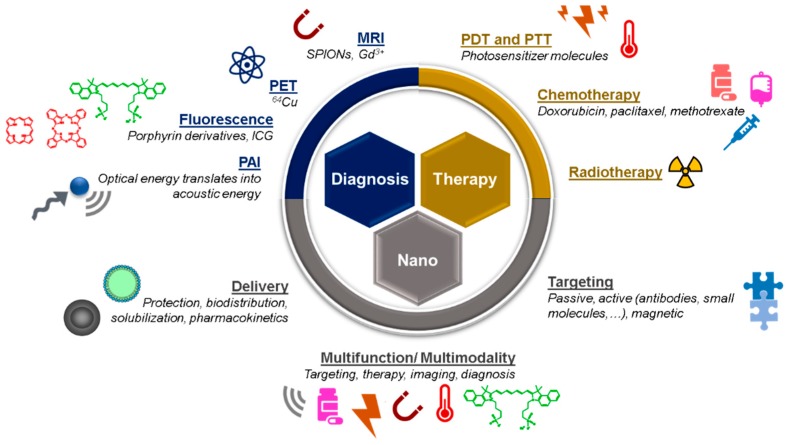
Schematic illustration of triple features in theranostic nanoplatforms, namely nano-sized particle, therapeutic, and diagnostic agents. Adapted with permission from Zhang P, Hu C, Ran W, Meng J, Yin Q, Li Y. Recent progress in light-triggered nanotheranostics for cancer treatment. Theranostics. 2016;6(7):948; copyright 2016 Ivyspring International Publisher [9].

**Figure 3 pharmaceutics-11-00022-f003:**
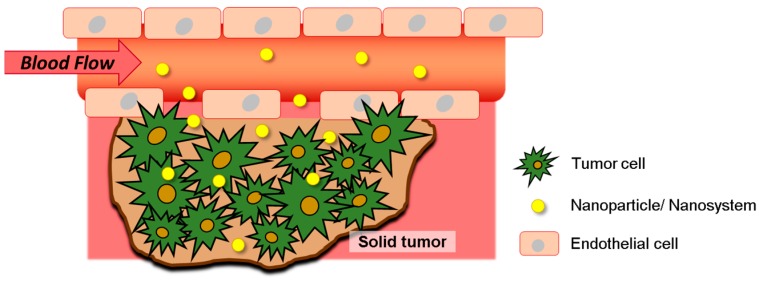
Illustration of the accumulation of nanosystems at tumor sites through an enhanced permeation and retention (EPR) effect.

**Figure 4 pharmaceutics-11-00022-f004:**
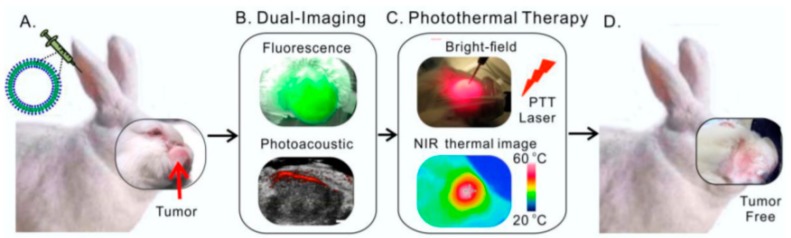
Schematic image of the dual-imaging and PTT effect on head and neck cancer from rabbit. (**A**) Porphysomes injected intravenously via ear vein of the rabbit; (**B**) at 24 h post-injection, both fluorescence and photoacoustic imaging were enabled; (**C**) in vivo photothermal ablation of rabbit tumor by two-step ablations (intra-tumor and transdermal); (**D**) the tumor was eliminated with no recurrence. Adapted with permission from Muhanna, Jin, Huynh, Chan, Qiu, Jiang, Cui, Burgess, Akens, Chen, and Irish. Phototheranostic porphyrin nanoparticles enable visualization and targeted treatment of head and neck cancer in clinically relevant models. Theranostics. 2015;5(12):1428; copyright 2015 Ivyspring International Publisher [74].

**Figure 5 pharmaceutics-11-00022-f005:**
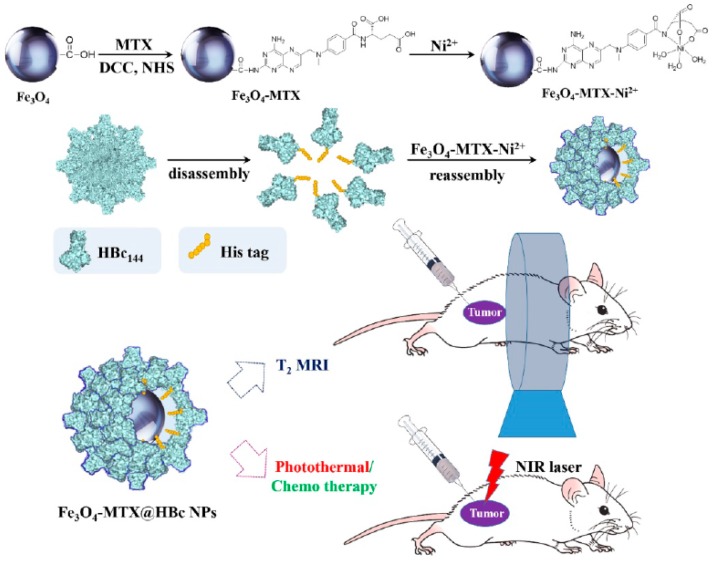
Schematic description of Fe_3_O_4_ magnetic nanoparticles conjugated with methotrexate for triple action in imaging (MRI contract agent) and therapeutic (photothermal and chemo therapy). Adapted with permission from Zhang, Shan, Ai, Chen, Zhou, Lv, Zhou, Ye, Ren, and Wang. Construction of Multifunctional Fe3O4-MTX@HBc Nanoparticles for MR Imaging and Photothermal Therapy/Chemotherapy, Nanotheranostics. 2018; 2(1): 87–95. doi: 10.7150/ntno.21942 copyright 2018 Ivyspring International Publisher [94].

**Table 1 pharmaceutics-11-00022-t001:** Theranostics polymeric and metal-based nanoplatforms under study for clinical applications in different cancers (according to clinicaltrials.gov and clinicaltrialsregister.eu).

Product	Company	Clinical Phase	Therapeutic Modality	Diagnostic Modality	Proposed Indication	CT Identifier
CriPec^®^ docetaxel	Cristal Therapeutics	Phase I	Docetaxel	PET (Zirconium-89)	Solid tumors	NCT03712423
AGuIX^®^	NHTherAguix	Phase I	Radiation therapy	MRI (gadolinium-chelates)	Brain metastases	NCT02820454
AGuIX^®^	NHTherAguix	Phase I	Radiation therapy or brachytherapy or chemotherapy (cisplatin)	MRI (gadolinium-chelates)	Gynecologic cancer	NCT03308604
Iron oxide nanoparticles (SPIONs)	M.D. Anderson Cancer Center	Early Phase I	-	Ferumoxytol-based MRI	HNSCC	NCT01895829
NBTXR3^®^	Nanobiotix	Phase I/II	Hafnium oxide nanoparticles (50 nm)	Radiation-stimulated technology (NanoX-Ray) via electron production	Multiple solid cancers, including head and neck cancer, rectal cancer, prostate cancer and breast cancer	NCT02805894 NCT03589339 NCT02901483 NCT02901483 NCT02465593 (Total of 5 active clinical trials)

Abbreviations: ANZUP: Australian and New Zealand Urogenital and Prostate Cancer Trials Group. HNSCC: head and neck squamous cell carcinoma. MRI: magnetic resonance imaging. PET: positron emission tomography. SPION: superparamagnetic iron oxide nanoparticles.

**Table 2 pharmaceutics-11-00022-t002:** Lipid-based nanotheranostic systems in a preclinical stage of development.

Lipid-Based System/Drug/Imaging Agent	Detection Method	Tumor	Animal Model	Observations	Reference
Thermosensitive liposomes/doxorubicin/Gd-DTPA-BMA	MRI	Soft Tissue Sarcoma	Brown Norway rat syngeneic model	Liposomal content release was promoted by local application of hyperthermia.	[14]
PEG liposomes (^14^C)/QDs	β Radiation; ICP-MS	Melanoma	C57BL/6 syngeneic model	Biodistribution study.	[119]
PEG liposomes/ruthenium polypyridine complex	Fluorescence	TNBC	Athymic nude mice orthotopic model	Ruthenium polypyridine complex was used for imaging and therapy.	[124]
PEG liposomes/PLP/Gd-DOTAMA(C_18_)_2_	MRI	Melanoma	C57BL/6 syngeneic model	-----	[125]
PEG liposomes/Gadoteridol	MRI	Melanoma	C57BL/6 syngeneic model	Liposomal content release was promoted by a local application of pLINFU.	[126]
Targeted PEG liposomes/ICG	PAI; US	Epidermoid Carcinoma	CD1 (nu/nu) xenograft model	Biodistribution study Riboflavin was used as a targeting moiety.	[127]
Targeted PEG liposomes/doxorubicin/ICG	MSOT	Breast; Colon	Athymic nude-Foxn1 mice xenograft model	Monoclonal antibody hCTM01 was used as a targeting moiety.	[128]
Thermosensitive PEG liposomes/ICG	NIR	TNBC	Athymic nude mice (nu/nu) xenograft model	PDT.	[129]
PEG liposomes/SPIONs	MRI	Breast	Transgenic mice - MMTV-PyMT	-----	[130]
PEG liposomes/AQ4N/photosensitizer *h*Ce6; ^64^Cu isotope	Fluorescence, PAI and PET	Breast	BALB/c syngeneic model	AQ4N is a hypoxia-activated prodrug.	[131]
PEG liposomes/doxorubicin/Gadoteridol	MRI	Breast	BALB/c syngeneic model	Liposomal content release was promoted by local application of pLINFU and/or sonoporation.	[132,133]
Targeted PEG liposomes/doxorubicin/fluorescent probe (PFBT)	Fluorescence	Breast	Nude mice xenograft model	Folate was used as a targeting moiety.	[134]
Targeted PEG liposomes/mitoxantrone/SPIONs	MRI	Breast	Athymic nude BALB/c xenograft model	Gonadorelin, a peptide analogue of luteinizing hormone-releasing hormone (LHRH), was used as a targeting moiety.	[135]
Targeted PEG liposomes/doxorubicin/NIR probe DiD	NIR	TNBC	Nude BALB/c xenograft model	A cyclic octapeptide was used as a targeting moiety.	[136]
PEG liposomes/doxorubicin, gemcitabine, cisplatin or caffeine/DNA barcode and ICG	NIR	TNBC	BALB/c syngeneic model	Specific DNA barcodes were used to screen the therapeutic potency of each anticancer drugs using tumor biopsies.	[137]
Temperature-sensitive PEG liposomes/doxorubicin/ICG	NIR	Breast	Nude BALB/c xenograft model	NIR laser-driven chemotherapy and PTT.	[138]
PEG liposomes/H_2_O_2_-dependent chromogenic reaction	NIR; PAI	Glioma; Breast; Lymph node metastasis	BALB/c orthotopic and syngeneic models	ABTS was used as substrate for HRP for chromogenic reaction; NIR was used for PTT.	[139]
Targeted pH-sensitive PEG liposomes/paclitaxel/SPIONs	MRI	Breast	Nude BALB/c xenograft model	The peptide H_7_K(R_2_)_2_ was used as the targeting moiety.	[140]
PEG liposomes/siRNA/Gd-DOTA-DSA	MRI	Ovary	Nude BALB/c xenograft model	Functional delivery of anti-survivin.	[141]
Targeted PEG liposomes/doxorubicin/ZnPc	Fluorescence; MRI	Cervical	Nude mice xenograft model	Methotrexate was used as an FR-targeting moiety.	[142]
Thermosensitive PEG liposomes/methotrexate; doxorubicin/fluorescent dye Cy5.5; iron oxide NPs	NIR; MRI	Cervical	Nude BALB/c xenograft model	Methotrexate used as an anticancer drug and an FR-targeting moiety.	[143]
PEG liposomes/oxaliplatin/SRB; Gd-DTPA	Fluorescence; MRI	Colon	BALB/c syngeneic model	Porphyrin-lipid was used for the light-triggered release of liposomal content.	[144]
Targeted PEG liposomes/HSV-TK/GCV suicide gene system/QDs	NIR	Liver	BALB/c (nu/nu) xenograft model	Folate was used as a targeting moiety.	[145]
PEG liposomes/H_2_S/SPIONs; DiD	MRI; US; NIR	Liver	Nude BALB/c xenograft model	H_2_S is a hydrogen sulfide prodrug; ADT was used as an organic H_2_S donor; SPIONs were used for magnetic targeting and imaging.	[146]
Targeted PEG liposomes/doxorubicin/Gd^3+^ texaphyrin	MRI; Fluorescence	Liver	BALB/c orthotopic model	Folate was used as a targeting moiety; doxorubicin was used as a therapeutic and imaging agent.	[147]
Thermosensitive PEG liposomes/IR820; Iohexol; Gd-DTPA	Fluorescence; MIR; CT	Glioma	Nude BALB/c xenograft model	IR820 was used as an imaging and PTT agent.	[148]
PEG liposomes/cilengitide/QDs; SPIONs	NIR; MIR	Glioblastoma	Sprague-Dawley rat orthotopic model	-----	[149]
Thermosensitive PEG liposomes/doxorubicin/[Gd(HPDO_3_A)(H_2_O)]	MRI	Gliosarcoma	Fisher 344 rat syngeneic model	Liposomal content release was promoted by a local application of HIFU.	[150]
Targeted PEG liposomes/Rituximab/SPIONs	MRI	Brain Lymphoma	Athymic nude mice xenograft model	Rituximab was used for targeting and therapy.	[151]
Targeted PEG lipid nanoparticles/porphyrin-lipid	Fluorescence	Glioblastoma	Athymic nude mice orthotopic model	apoE3 was used as a targeting moiety and porphyrin-lipid as an imaging modality.	[152]
Targeted SLNs/hydrophobic IR780 dye	NIR	Glioblastoma	Athymic nude mice xenograft model	Cycle RGD peptide (cRGD) was used as a targeting moiety and IR780 as an imaging and PTT agent.	[153]
Targeted NLCs/NIR dye IR780	NIR	Breast	BALB/c syngeneic model	NIR dye IR780 was used for imaging and PTT.	[154]
NLCs/paclitaxel/QDs	NIR	Liver	Kunming mice syngeneic model	-----	[155]
Targeted nanocapsules/paclitaxel/DiD	Fluorescence	Glioblastoma	C57Bl/6 orthotopic model	NFL-TBS.40-63 (cell-penetrating peptide) was used as a targeting moiety.	[156]
Micelles/docetaxel/NIR probe DiR	NIR	Breast	Nude BALB/c xenograft model	-----	[157]

PEG: 1,2-distearoyl-sn-glycero-3-phosphoethanolamine-*N*-[amino(polyethylene glycol)] (DSPE-PEG); QDs: quantum dots; ICP-MS: inductively coupled plasma mass spectrometry; PLP: prednisolone phosphate; Gd: gadolinium; MRI: magnetic resonance imaging; pLINFU: pulsed low-intensity non-focused ultrasound; NIR: near infrared; DiR: lipophilic fluorochrome (1,1-dioctadecyl-3,3,3,3-tetramethylindotricarbocyanine iodide); SPIONs: superparamagnetic nanoparticles; PFBT: poly(9,9-dioctylfluorene-2,7-diyl-co-benzothiadiazole); NLCs: nanostructured lipid carriers; PTT: photothermal therapy; LHRH: luteinizing hormone-releasing hormone; *h*Ce6: hexadecylamine conjugated chlorin e6; PAI: photoacoustic imaging; PET: positron emission tomography; ICG: indocyanine green; MSOT: multispectral optoacoustic tomography; hCTM01: anti-MUC- 1 “humanized” monoclonal antibody (MoAb); DNA: deoxyribonucleic acid; TNBC: triple negative breast cancer; PDT: photodynamic therapy; DiD: lipophilic fluorochrome (1,1-Dioctadecyl-3,3,3,3-tetramethylindodicarbocyanine); siRNA: small interfering ribonucleic acid (RNA); Cy5.5: fluorescent dye; NPs: nanoparticles; FR: folate receptor; ZnPc: zinc phthalocyanine; apoE3: apolipoprotein E3; NFL-TBS.40-63: neurofilament derived cell-penetrating peptide; SLNs: solid lipid nanoparticles; Gd-DTPA: gadolinium-diethylenetriamine pentaacetic acid; CT: computerized tomography; HIFU: high intensity focused ultrasound; HSV-TK-GCV: herpes simplex virus thymidine kinase gene (HSV-TK) with ganciclovir (GCV); H_2_S = hydrogen sulfide prodrug; US: ultrasound; ADT: anethole dithiolethione; NLCs: nanostructured lipid carriers; SRB: sulforhodamine B; H_2_O_2_: hydrogen peroxide; HRP: horseradish peroxidase; ABTS: 2,2′-azino-bis(3-ethylbenzothiazoline-6-sulfonic acid); Gd-DTPA-BMA: gadolinium-diethylenetriamine pentaacetic acid-bis methylamine.

**Table 3 pharmaceutics-11-00022-t003:** Liposomal nanotheranostics in clinical studies for application in different malignancies.

Lipid-Based System/Drug/Imaging Agent	Detection Method	Cancer	Clinical Phase	CT Identifier	Reference
PEG liposomes/doxorubicin/^99m^Tc	SPECT/CT	Ovary	Early Study	N.A.	[182]
HER2-targeted PEG liposomes/doxorubicin/^64^Cu	PET	Advanced Breast Cancer	Phase I	NCT01304797	[183]
Lyso-thermosensitive liposomes (TARDOX)/doxorubicin/US	US	Liver	Phase I	NCT02181075	[184]

^99m^Tc: Technetium 99m; SPECT: single-photon emission computed tomography; CT: computerized tomography; PET: positron emission tomography; US: ultrasound.

**Table 4 pharmaceutics-11-00022-t004:** The performance of metallic, polymeric, and lipid-based nanosystems as cancer nanotheranostics.

	Type	Polymeric	Metallic *	Lipid-based
Features				
Preparation method (complexity)		++	+	++
Physico-chemical characterization (easiness)		++	+	+++
Stability		+++	+	++
Multifunctionality (possibility to apply different external stimuli)		++	+++	++
Potential toxicity		++	+++	+
In vivo general performance		++	+	+++
Scale-up (easiness)		+	++	+
Cost		++	+	+

Classification: + low; ++ moderate; +++ high. (*) Focused on gold-nanoparticles based nanotheranostics.

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
