# Peer review of "Current Trends in Cancer Nanotheranostics: Metallic, Polymeric, and Lipid-Based Systems"

_pharmaceutics, 2019, doi:10.3390/pharmaceutics11010022_

Round 1
Reviewer 1 Report
This review article gives us a detailed overview of development of metallic, polymeric and lipid-based nano formulations in theranostic of solid cancers, including kin cancer, head and neck, thyroid, breast, gynecologic, prostate, colon cancers, brain-related cancer and hepatocellular carcinoma. I would like to recommend publication after the authors address the following two minor points.
1. Section 2.2 is more about taking advantage of tumor microenvironment characteristics to target cancer cells, not targeting tumor microenvironment. The subtitle is misleading.
2. In section 2.3, the authors are recommended to comment on the reason why no current nanotheranostic formulation has been approved in clinic. I suggest the authors to take a deeper look at nanotheranostic platforms under clinical trail and discuss the requirements or criteria nanotheranostic platforms have to meet for translation.
Author Response
Response to Reviewer 1 Comments
1. Section 2.2 is more about taking advantage of tumor microenvironment characteristics to target cancer cells, not targeting tumor microenvironment. The subtitle is misleading.
Author Response
Thank you for your suggestion. The subtitle of section 2.2 “Response to the needs in targeting tumor microenvironment” was changed to “Exploring tumor microenvironment for improved nanotheranostics targeting”
2. In section 2.3, the authors are recommended to comment on the reason why no current nanotheranostic formulation has been approved in clinic. I suggest the authors to take a deeper look at nanotheranostic platforms under clinical trial and discuss the requirements or criteria nanotheranostic platforms have to meet for translation.
Author Response
We appreciate the referee’s comment. A new section was included in the revised manuscript before conclusions: Section “4. Meeting the criteria for a successful translation of nanotheranostics into clinic settings”.
In this section the requirements that nanotheranostics platforms have to meet for translation were discussed. This section is highlighted in blue in the revised manuscript.
Reviewer 2 Report
In this manuscript, Silva et al. reviewed recent progress in metallic, polymeric, and lipid-based nanosystems for cancer theranostics. The manuscript is interesting to read and can provide and timely and useful review to the field. The authors can improve the manuscript according to the following comments.
1. A table comparing the performance of metallic, polymeric, and lipid-based nanosystems in terms of cancer nanotheranostics should be added.
2. PLGA based hybrid polymeric nanoparticles are an important class of nanocarriers for cancer treatment. The authors should briefly elaborate this point by citing references such as Nanomedicine 2017, 13(8): 2451-2462; Molecular Pharmaceutics 2017, 14(8): 2697-2710; Colloids Surf B Biointerfaces 2017, 159: 217-231; ACS Omega 2018, 3(8): 9210-9219 et al.
Author Response
Response to Reviewer 2 Comments
1. A table comparing the performance of metallic, polymeric, and lipid-based nanosystems in terms of cancer nanotheranostics should be added.
Author response:
According to referee’s comment, the Table 4 “Comparison between the performance of metallic, polymeric, and lipid-based nanosystems as cancer nanotheranostics “was included in the revised manuscript. In this table different nanotheranostics reviewed in the manuscript were compared in terms of key criteria for assessing their potential translation into clinical settings.
2. PLGA based hybrid polymeric nanoparticles are an important class of nanocarriers for cancer treatment. The authors should briefly elaborate this point by citing references such as Nanomedicine 2017, 13(8): 2451-2462; Molecular Pharmaceutics 2017, 14(8): 2697-2710; Colloids Surf B Biointerfaces 2017, 159: 217-231; ACS Omega 2018, 3(8): 9210-9219 et al.
Author response
Thank you for your suggestion, the recommended references were included in the revised manuscript. They correspond to references nº 20, 21, 22 and 23. References nº 24 and 25 were also added in the last version of the manuscript.
Round 2
Reviewer 2 Report
The authors have addressed my comments.